# A conserved quality-control pathway that mediates degradation of unassembled ribosomal proteins

**Min-Kyung Sung[1], Tanya R Porras-Yakushi[2], Justin M Reitsma[1], Ferdinand M Huber[3], Michael J Sweredoski[2], André Hoelz[3], Sonja Hess[2], Raymond J Deshaies[1,4]\***

[1]Division of Biology and Biological Engineering, California Institute of Technology, Pasadena, United States; [2]Proteome Exploration Laboratory, Division of Biology and Biological Engineering, Beckman Institue, California Institute of Technology, Pasadena, United States; [3]Division of Chemistry and Chemical Engineering, California Institute of Technology, Pasadena, United States; [4]Howard Hughes Medical Institute, California Institute of Technology, Pasadena, United States

**Abstract** Overproduced yeast ribosomal protein (RP) Rpl26 fails to assemble into ribosomes and is degraded in the nucleus/nucleolus by a ubiquitin-proteasome system quality control pathway comprising the E2 enzymes Ubc4/Ubc5 and the ubiquitin ligase Tom1. *tom1* cells show reduced ubiquitination of multiple RPs, exceptional accumulation of detergent-insoluble proteins including multiple RPs, and hypersensitivity to imbalances in production of RPs and rRNA, indicative of a profound perturbation to proteostasis. Tom1 directly ubiquitinates unassembled RPs primarily via residues that are concealed in mature ribosomes. Together, these data point to an important role for Tom1 in normal physiology and prompt us to refer to this pathway as ERISQ, for excess ribosomal protein quality control. A similar pathway, mediated by the Tom1 homolog Huwe1, restricts accumulation of overexpressed hRpl26 in human cells. We propose that ERISQ is a key element of the quality control machinery that sustains protein homeostasis and cellular fitness in eukaryotes.

*For correspondence: deshaies@ caltech.edu

## Introduction

Protein quality control (PQC) has emerged as a major mechanism for maintaining protein homeostasis and cellular fitness. Defects in the cellular machinery that governs PQC cause multiple human diseases including multisystem proteinopathy (*Brandmeir et al., 2008*; *Watts et al., 2004*) and Amyotrophic Lateral Sclerosis (ALS) (*Johnson et al., 2010*; *Kabashi and Durham, 2006*). Other diseases, such as cancer, can exhibit heightened dependency on PQC pathways, which underlies the hypersensitivity of multiple myeloma cells to proteasome inhibitors (*Cenci et al., 2012*; *Deshaies, 2014*). Therefore, a deeper understanding of PQC will advance our understanding of both normal physiology and pathological states, and may enable novel approaches to treat multiple diseases.

Ribosome biogenesis is an intricate process involving many chaperones and assembly factors (*Kressler et al., 2010*; *Warner, 1999*). Ribosomal proteins made in excess over rRNA and other ribosomal proteins are among the most rapidly degraded proteins in eukaryotic cells (*Abovich et al., 1985*; *Dephoure et al., 2014*; *Gorenstein and Warner, 1977*; *Torres et al., 2010*, *2007*; *Warner, 1977*), suggesting that proper coordination of synthesis and assembly is critical. Newly-synthesized human ribosomal proteins are subject to degradation by the proteasome in the

**eLife digest** Ribosomes are the molecular machines in cells that produce proteins. The ribosomes themselves are composed of almost 80 different proteins that are held together by scaffolds made from molecules of RNA. Each protein is present in one copy, and so equal numbers of all proteins are needed to assemble a ribosome. However, because it takes many steps to produce a protein and biological processes are inherently imprecise, it is essentially impossible for a cell to produce exactly the same number of copies of all the proteins in a ribosome. Much research suggests that, to overcome these issues, a cell will make more of certain ribosomal proteins than it needs, and then degrade the leftovers that are not used. However, it was not clear how this happens, nor was it known what are the consequences of failing to degrade the leftovers.

Now, Sung et al. show that yeast cells use an enzyme named Tom1 to attach a protein-marker called ubiquitin to ribosomal proteins that are made in excess and not assembled into ribosomes. The ubiquitin serves as a tag that marks proteins for degradation, and yeast cell that lack Tom1 fail to degrade any excess ribosomal proteins. Consequently, the mutant yeast become sensitive to any factors that alter the balance of the protein and RNA building blocks used to assemble ribosomes.

The human equivalent of Tom1 is known as Huwe1, and the data of Sung et al. suggest that this enzyme acts in a similar pathway. Further experiments are now needed to explore the role of Huwe1 in greater depth, and investigate if problems with this enzyme are associated with any human diseases. Finally, working out the exactly how Tom1 recognizes unassembled ribosomal proteins will be another important challenge for future studies.

nucleolus (*Lam et al., 2007*), and we recently found that overexpressed yeast ribosomal proteins that fail to assemble are conjugated with ubiquitin and degraded by the proteasome in the nucleus (*Sung et al., 2016*). Insoluble material that accumulates upon transient inhibition of the proteasome in yeast is strongly enriched for ribosomal proteins (*Sung et al., 2016*), pointing to PQC of unassembled ribosomal proteins as a major pathway of proteostasis. However, the PQC pathway that mediates ERISQ remains unknown – an important gap in our understanding of PQC that we set out to address.

## Results

### Identification of Ubc4/5 and Tom1 as the E2 and E3 for ERISQ

We evaluated 115 mutant yeast strains, each lacking a different non-essential ubiquitin-proteasome system (UPS) gene, for those that accumulated non-essential ribosomal protein Rpl26a tagged with a FLAG epitope (Rpl26a$^{FLAG}$) upon its overexpression from the *GAL10* promoter. Accumulation of Rpl26a$^{FLAG}$ in most mutants was similar to wild type (WT) and well below the level detected in *rpl26aΔrpl26bΔ* (*Figure 1—figure supplement 1A and B*), which accumulated overexpressed Rpl26a$^{FLAG}$ due to lack of competition from endogenous Rpl26 (*Sung et al., 2016*). Notably, Rpl26a$^{FLAG}$ accumulated to high levels in *tom1Δ* and *ubc4Δ* cells (*Figure 1A* and *Figure 1—figure supplement 1A and B*).

Ubc4 is an ubiquitin-conjugating enzyme (E2) that is paralogous to and functionally redundant with Ubc5 (*Seufert and Jentsch, 1990*). Thus, subsequent experiments were performed with *ubc4Δubc5Δ* mutants. To test whether Ubc4/Ubc5 promoted ubiquitination of unassembled ribosomal proteins, we examined ubiquitin conjugates of overexpressed Rpl26a$^{FLAG}$ that accumulated in proteasome-deficient *pre9Δ* cells (*Sung et al., 2016*). Ubiquitinated Rpl26a$^{FLAG}$ was detected in *pre9Δ* but not in *ubc4Δubc5Δpre9Δ* cells (*Figure 1B*), indicating that Ubc4/Ubc5 promote ubiquitination of excess Rpl26a.

Tom1 is an E3 ubiquitin ligase of the HECT (homologous to E6AP C terminus) family. To investigate Tom1 function, we constructed *tom1$^{CA}$* strains in which the endogenous *TOM1* locus was mutated such that the catalytic cysteine3235 was changed to alanine (*tom1$^{CA}$*). We also appended a 3×HA epitope sequence to the 5' end of both *TOM1* and *tom1$^{CA}$*, and confirmed that the $^{3×HA}$Tom1 and $^{3×HA}$Tom1$^{CA}$ proteins were expressed equivalently (*Figure 1—figure supplement 2A*) and the

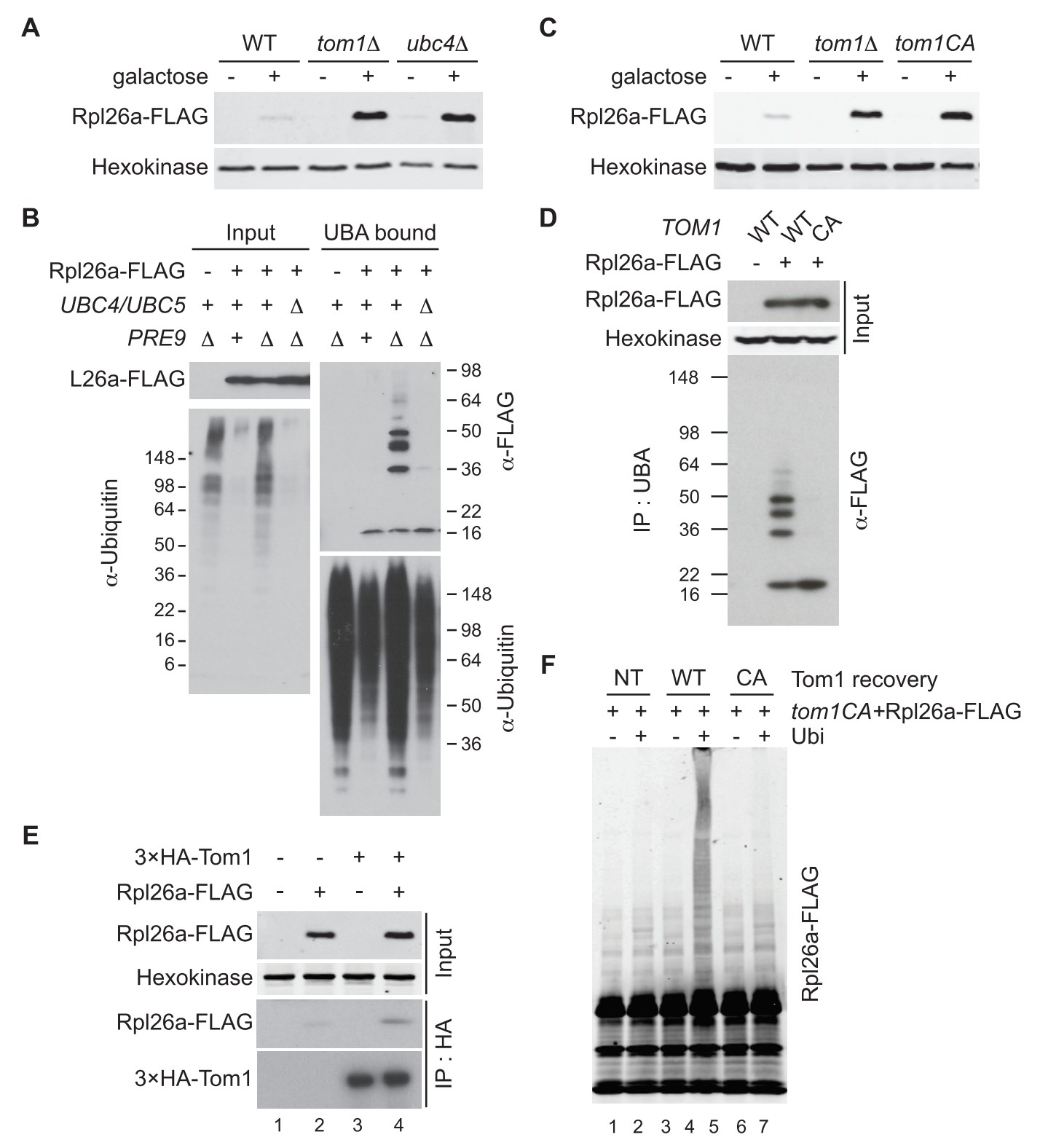

**Figure 1.** Ubc4/5 and Tom1 are the E2 and E3 enzymes responsible for ERISQ. (**A**) Rpl26a[FLAG] accumulates in *tom1Δ* and *ubc4Δ*. Accumulation of Rpl26a[FLAG] upon galactose induction in WT, *tom1Δ and ubc4Δ* cells was evaluated by SDS-PAGE and immunoblotting with the indicated antibodies. n = 3 biological replicates. (**B**) Rpl26a[FLAG] ubiquitination depends on Ubc4/Ubc5. Rpl26a[FLAG] was induced in cells of the indicated genotypes and cell lysates were prepared and subjected to pull-down with UBA domain resin. Input and bound proteins were evaluated as in (**A**). n = 3 biological

*Figure 1 continued on next page*

*Figure 1 continued*

replicates. (C) Rpl26a$^{FLAG}$ accumulates in *tom1$^{CA}$* cells. As in (A) except that the Tom1 ligase-dead (*tom1$^{CA}$*) mutant was used. n = 3 biological replicates. (D) Rpl26a$^{FLAG}$ ubiquitination depends on Tom1. As in (B) except that cells expressing WT Tom1 or Tom1$^{CA}$ were treated with bortezomib for 45 min after addition of galactose. n = 3 biological replicates. (E) Rpl26a$^{FLAG}$ binds $^{3\times HA}$Tom1. Anti-HA immunoprecipitates from cells expressing $^{3\times HA}$Tom1 and Rpl26a$^{FLAG}$ were immunoblotted with antibodies to HA, FLAG, and hexokinase. n = 3 biological replicates. (F) In vitro ubiquitination of Rpl26a$^{FLAG}$ by Tom1. Rpl26a$^{FLAG}$ retrieved in $^{3\times HA}$Tom1$^{CA}$ immunoprecipitates was supplemented or not with E1/E2/ubiquitin/ATP (Ubi) and Tom1 retrieved from untagged (NT), $^{3\times HA}TOM1$ (WT), or $^{3\times HA}TOM1^{CA}$ (CA) cells, as indicated. See detailed methods in Material and methods. n = 3 biological replicates.

The following figure supplements are available for figure 1:

**Figure supplement 1.** Identification of ERISQ defect in *tom1Δ* and *ubc4Δ*.

**Figure supplement 2.** Characterization of tagged and ligase-dead Tom1.

**Figure supplement 3.** Tom1 mediates ubiquitination of overexpressed Rpl26a.

$^{3\times HA}$Tom1 was functional (*Figure 1—figure supplements 2B–D*). Using these strains, we established that Tom1 E3 activity was required for repression (*Figure 1C*) and ubiquitination (*Figure 1D* and *Figure 1—figure supplement 3A*) of overexpressed Rpl26a$^{FLAG}$. Rpl26a$^{FLAG}$ was co-immunoprecipitated with $^{3\times HA}$Tom1 (*Figure 1E*). Upon addition of an ubiquitination cocktail, immunoprecipitations of wild type but not mutant $^{3\times HA}$Tom1 ubiquitinated co-precipitated Rpl26a$^{FLAG}$ (*Figure 1—figure supplement 3B*). Importantly, the activity defect of the $^{3\times HA}$Tom1$^{CA}$ immunoprecipitate was complemented by adding $^{3\times HA}$Tom1 but not $^{3\times HA}$Tom1$^{CA}$ prior to the ubiquitination reaction (*Figure 1F* and *Figure 1—figure supplement 3C*).

To identify the population of Rpl26a$^{FLAG}$ targeted by Tom1, we performed sucrose gradient fractionation. Mutant *tom1$^{CA}$* cells, like cells treated with the proteasome inhibitor bortezomib (*Sung et al., 2016*), accumulated unassembled Rpl26a$^{FLAG}$ that co-fractionated with $^{3\times HA}$Tom1$^{CA}$ (*Figure 2A*; note that $^{3\times HA}$Tom1 and $^{3\times HA}$Tom1$^{CA}$ fractionated similarly). Co-immunoprecipitation of $^{3\times HA}$Tom1 or $^{3\times HA}$Tom1$^{CA}$ with Rpl26a$^{FLAG}$ was only detected in these low MW fractions (*Figure 2B*). Moreover, ubiquitinated Rpl26a$^{FLAG}$ detected in low MW fractions from bortezomib-treated cells was almost entirely lost from *tom1$^{CA}$* cells (*Figure 2B*). Consistent with the reported localization of Tom1 (*Huh et al., 2003*), Rpl26a$^{FLAG}$ or Rpl26a$^{GFP}$ that accumulated upon their transient overexpression in *tom1$^{CA}$* cells were found in the nucleus and nucleolus (*Figure 2C*). Taken together, these data provide strong evidence that overexpressed Rpl26a failed to assemble into ribosomes and was directly bound and ubiquitinated by Tom1 in the nuclear/nucleolar compartments.

## Tom1 targets a broad range of ribosomal proteins

To address whether Tom1 might have a broader role in promoting degradation of excess ribosomal proteins other than Rpl26a, we evaluated accumulation of a set of eight ectopically overexpressed ribosomal proteins in *tom1Δ* and WT cells. Similar to what we observed with bortezomib (*Sung et al., 2016*), deletion of *TOM1* enabled increased accumulation of at least seven of them (*Figure 3—figure supplement 1A*). We next sought to test whether Tom1 promoted degradation of unassembled ribosomal proteins in cells in which they were not deliberately overexpressed. We reasoned that if this is the case, Tom1 should directly associate with ribosomal proteins. Mass spectrometry of $^{3\times HA}$Tom1 immunoprecipitates from bortezomib-treated cells revealed enrichment for several ribosomal proteins, including Rpl26b (*Figure 3—figure supplement 1B* and *Supplementary file 3A*).

Ribosomal proteins are commonly identified in purified ubiquitin conjugates (*Mayor et al., 2007*, *2005*; *Peng et al., 2003*) or in ubiquitination site mapping experiments that rely on purification of the GlyGly dipeptide that remains attached to a lysine side chain following digestion of an ubiquitin conjugate with trypsin (*Kim et al., 2011*; *Lesmantavicius et al., 2014*; *Porras-Yakushi and Hess, 2014*; *Porras-Yakushi et al., 2015*; *Sarraf et al., 2013*; *Swaney et al., 2013*; *Udeshi et al., 2013b*; *Wagner et al., 2011*). Thus, we reasoned that if Tom1 plays a broad role in PQC of unassembled

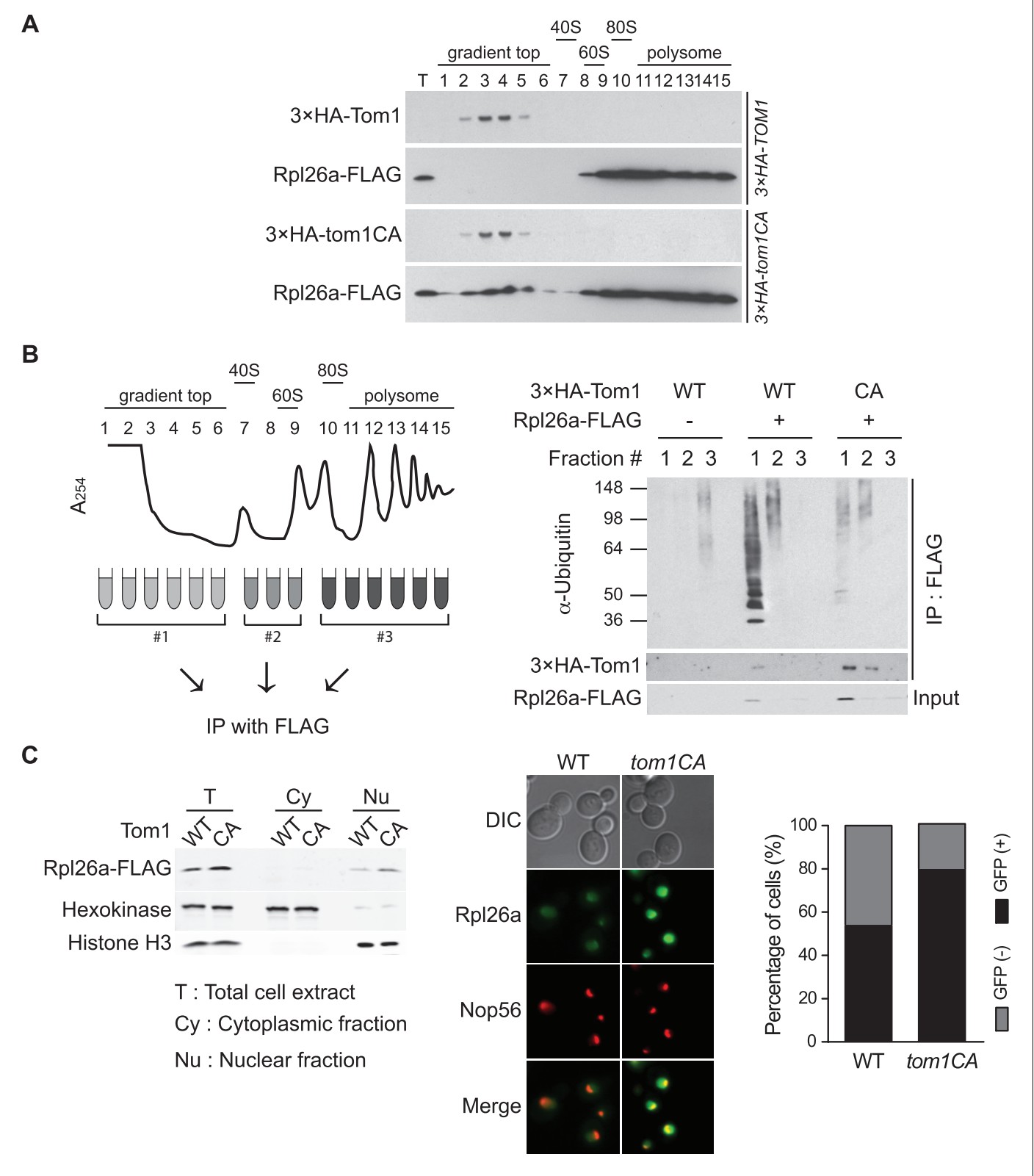

**Figure 2.** Tom1 functions in non-ribosomal fractions. (**A**) Sucrose gradient fractionation behavior of [3xHA]Tom1 and Rpl26a[FLAG] upon galactose induction of Rpl26a[FLAG] in [3×HA]TOM1 or [3×HA]TOM1[CA] cells. T indicates total extract. n = 2 biological replicates. (**B**) Tom1 is required for ubiquitination of unassembled Rpl26a[FLAG]. Left: experimental scheme. Right: cells were treated with bortezomib for 30 min after induction of Rpl26a[FLAG] with galactose and then lysed and fractionated as in panel A prior to being processed as depicted in panel B. n = 2 biological replicates. (**C**) Rpl26a accumulates in

*Figure 2 continued on next page*

*Figure 2 continued*

the nucleus of *tom1^CA* cells. Left: Subcellular fractionation of Rpl26a^FLAG induced in WT and *tom1^CA* cells. Histone H3 and Hexokinase were used as nuclear and cytoplasmic markers, respectively. CA refers to Tom1-Cys3235Ala. Right: Fluorescence microscopy of Rpl26a^GFP induced in WT and *tom1^CA* cells. Nop56-RFP marks nucleoli. Shown at far right is the percentage of GFP positive cells. n = 2 biological replicates.

ribosomal proteins as suggested by the experiments shown in *Figure 3—figure supplement 1A and B*, perhaps it accounts for the frequent recovery of ribosomal proteins in prior global ubiquitin conjugate profiling efforts. To address this possibility, we performed quantitative GlyGly profiling of *tom1Δ* and *TOM1* cells using SILAC (*Figure 3—figure supplement 1C*) to identify changes in the level of ubiquitination of specific lysines that occur upon loss of Tom1. Analysis of three biological replicates (*Figure 3A* and *Figure 3—figure supplement 2A* and *Supplementary file 3B*) revealed 1980 unique ubiquitination sites in 920 distinct proteins, of which 972 unique sites in 532 proteins were quantified. All three *tom1Δ* biological replicates exhibited lower overall ubiquitination than wild type, suggesting a major role for Tom1 in PQC. Of the 141 sites that exhibited a $\geq$two-fold decrease in ubiquitination in *tom1Δ*, 51 (36%) were in ribosomal proteins (*Figure 3—figure supplement 2B*). Moreover, of the ubiquitinated peptides derived from ribosomal proteins, >50% (51 of 101) decreased in abundance in *tom1Δ*. By comparison, of 837 non-ribosomal sites identified, only 11% decreased in abundance in *tom1Δ*. SILAC analysis of the unfractionated cell lysates indicated that the reduction in ribosomal ubiquitin conjugates in *tom1Δ* was not due to reduction in total ribosomal protein levels (*Figure 3B*). Gene ontology analysis of the GlyGly profiling data confirmed that ubiquitination of ribosomal proteins (*Figure 3C*), particularly those of the large (60S) subunit (*Figure 3D*), was disproportionately impacted by loss of Tom1. These trends are clearly evident from a plot of the top 25 Tom1-dependent modification sites within all proteins (*Figure 3—figure supplement 2C*) or just ribosomal proteins (*Figure 3E*). To address whether the strong effects on ribosomal protein ubiquitination seen in *tom1Δ* cells were due specifically to loss of Tom1's E3 activity, the GlyGly SILAC analysis was repeated with WT and *tom1^CA* cells. Quantitative analysis of the data confirmed a disproportionate loss of ribosomal protein ubiquitination (*Figure 3—figure supplement 2D*).

## Endogenous ribosomal proteins aggregate in *tom1Δ* mutants

If Tom1 mediates degradation of unassembled ribosomal proteins in unperturbed cells, there should not only be a decrease in ribosomal ubiquitin conjugates in *tom1* mutants, but a commensurate increase in unassembled ribosomal proteins that fail to be degraded. Since preliminary sucrose gradient fractionations did not reveal high levels of unassembled ribosomal proteins in *tom1Δ* cells (data not shown), we reasoned that over time, undegraded excess ribosomal proteins might aggregate and collect in insoluble deposits. To investigate this matter, we prepared detergent-insoluble fractions from WT cells treated with or without bortezomib (btz) and *tom1Δ* cells, and evaluated them for their content of ribosomal proteins. Detergent-insoluble proteins, including Rpl1 and Rpl3, were greatly increased in *tom1Δ* cells compared to WT cells (*Figure 3F*, Pellet). High accumulation of insoluble proteins in *tom1Δ* cells was evident regardless of the method or buffer employed for cell lysis (*Figure 3—figure supplement 3A*). This observation was confirmed and extended by mass spectrometry coupled with label-free absolute quantification using iBAQ (intensity-Based Absolute Quantification) (*Geiger et al., 2012*). The insoluble proteins that exhibited the largest increase in *tom1Δ* cells were ribosomal proteins including Rpl1 and Rpl3 (*Figure 3G*, *Figure 3—figure supplement 3B*, and *Supplementary file 3C*). Gene ontology analysis (*Figure 3H*) and a plot of the top 20 detergent-insoluble proteins (*Figure 3—figure supplement 3C*) indicated that ribosomal proteins, including those of both the 60S and 40S subunits (*Figure 3—figure supplement 3D*) comprise the major class of aggregating proteins in *tom1Δ* cells.

## Tom1 works through residues that are normally inaccessible in the structure of the mature ribosome

GlyGly profiling and analysis of insoluble proteins in *tom1* mutants both pointed to a broad role of Tom1 in ubiquitinating and degrading excess, unassembled ribosomal proteins. This raised a critical question that is common to all PQC pathways yet is poorly understood: how does Tom1 ubiquitinate

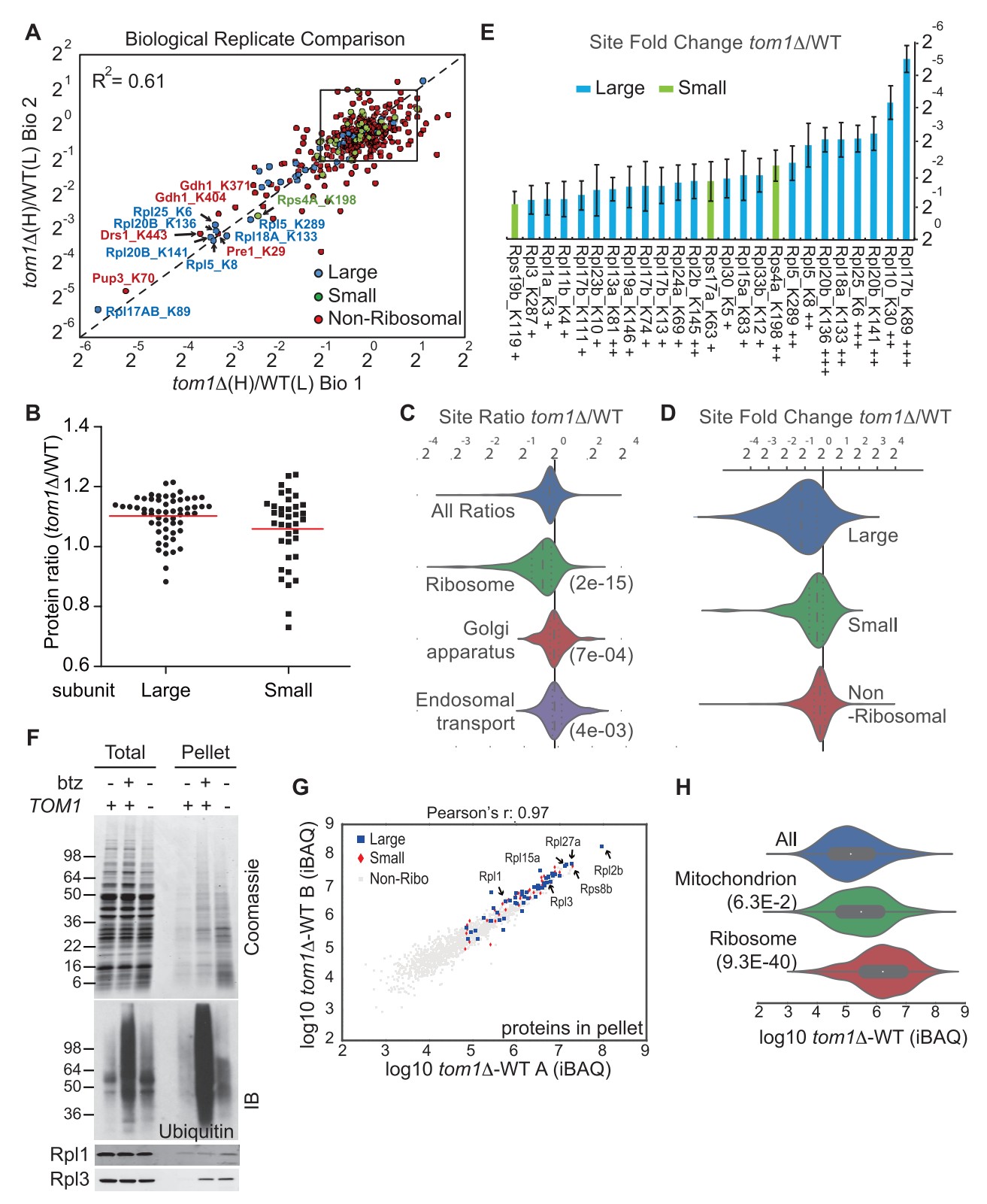

**Figure 3.** Diminished ubiquitination and accumulation of insoluble ribosomal proteins in *tom1* cells. (**A**) Diminished ubiquitination of ribosomal proteins in *tom1Δ*. Scatter plot of the SILAC ratios (*tom1Δ*/WT) for GlyGly-modified peptides identified in biological replicate 1 versus 2. Sites with the largest decrease in ubiquitination are annotated. The other pairwise comparisons are in *Figure 3—figure supplement 2A*. n = 3 biological replicates. (**B**) Column scatter plot representing the distribution of ratios (*tom1Δ*/WT) for proteins of the large (circles) and small (square) ribosomal subunits. A red bar

*Figure 3 continued on next page*

*Figure 3 continued*

indicates the mean. (C) Violin plot of gene ontology analysis of ubiquitinated proteins that had one or more ubiquitination site that decreased by $\geq$ 2-fold. The most strongly affected categories are shown. The number in parentheses refers to the disproportionate enrichment for the category in the top 10% of identifications and is the Benjamini and Hochberg corrected p-value from a Fisher Exact test. (D) Violin plot representing the distribution of ubiquitin site occupancy ratios (*tom1Δ*/WT) for the large (blue) and small (green) ribosome subunits, and non-ribosomal proteins (red). (E) The 25 ribosomal ubiquitination sites with the largest decrease in ubiquitin occupancy in *tom1Δ*. +++p<0.001; ++p<0.01; +p<0.05. Each site was observed in at least two of the three biological replicates. The error bars represent 95% confidence intervals. Note that ubiquitination at K37 and K69 in Rpl26b was decreased by 2.4-fold and 1.6-fold in *tom1Δ*, respectively (*Supplementary file 3B*). (F, G) Insoluble ribosomal proteins accumulate in *tom1Δ*. (F) Detergent-insoluble pellet fractions isolated from lysate (Total) of indicated cells were analyzed by SDS-PAGE and staining with Coomassie blue or immunoblotting with the indicated antibodies. The pellet fraction is overloaded 20-fold compared to the total and supernatant fractions. n = 2 biological replicates. (G) Scatter plot representing ΔiBAQ of biological replicate B *vs*. A for insoluble proteins in *tom1Δ* mutants. Ribosomal proteins with the largest increase in the pellet fraction upon *TOM1* deletion, and Rpl1 and Rpl3 are annotated. Pearson's r-value is indicated on top of the plot. The other pairwise comparisons are in *Figure 3—figure supplement 3B*. n = 3 biological replicates. (H) Gene ontology analysis of proteins exhibiting increased insolubility in *tom1Δ*. Analysis is the same as for panel C.

The following figure supplements are available for figure 3:

**Figure supplement 1.** Tom1 targets a broad range of overexpressed and endogenous ribosomal proteins.

**Figure supplement 2.** Quantitative GlyGly proteomic analyses of *tom1* mutants.

**Figure supplement 3.** Endogenous ribosomal proteins accumulate as insoluble species in *tom1Δ* mutants.

so many different ribosomal proteins, yet manage to maintain some level of specificity for unassembled forms? To begin to address this question, we constructed a mutant Rpl26a that did not bind to rRNA. Two positively-charged clusters in Rpl26 – RRKARK (amino acids 12–17) and a patch formed by R27, R28, R51, and R52 – mediate binding to 5.8S rRNA and assembly into ribosomes (*Babiano et al., 2012*) (*Figure 4A*). We mutated various combinations of these residues to glutamate and observed that some mutants exhibited even less accumulation than overexpressed WT Rpl26a-$^{FLAG}$ (*Figure 4B*).

Unlike WT Rpl26a$^{FLAG}$, Rpl26-4E$^{FLAG}$(R12E, R13E, R16E, K17E) did not accumulate or assemble into ribosomes in *rpl26aΔrpl26bΔ* cells (*Figure 4C*, top panels). However, upon inhibition of the proteasome with MG132 or bortezomib, Rpl26a-4E$^{FLAG}$ accumulated (*Figure 4—figure supplement 1A*) and was detected in the low MW fractions of a sucrose gradient (*Figure 4C*, bottom panels), where it was ubiquitinated, albeit to a lesser extent than unassembled WT Rpl26a$^{FLAG}$ (*Figure 4D*). Strikingly, Rpl26a-4E$^{FLAG}$ exhibited poor association with $^{3xHA}$Tom1 (*Figure 4E*) and showed only very weak accumulation in *tom1Δ* (*Figure 4F*), suggesting that it is not primarily a Tom1 substrate but is re-directed to another QC pathway. Consistent with this idea, Rpl26a-4E$^{FLAG}$ accumulated in *doa10* mutants (*Figure 4—figure supplement 1B*). The failure of Rpl26a-4E$^{FLAG}$ to be targeted by Tom1 was not due to a defect in its nuclear localization (*Figure 4G* and *Figure 4—figure supplement 1C*). We suggest that upon its import into the nucleus, Rpl26a-4E becomes a substrate for Doa10 that is localized to the inner nuclear membrane (*Deng and Hochstrasser, 2006*). Taken together, our results suggest that residues of Rpl26a that mediate interactions with rRNA and are buried when incorporated into the ribosome are also required for ubiquitination by Tom1 when Rpl26a fails to assemble.

To pursue this observation further and probe its potential generality, we focused on Rpl4, because assembly of Rpl4 has been studied in some depth (*Stelter et al., 2015*). Upon its synthesis, Rpl4 binds the dedicated chaperone Acl4. The Rpl4–Acl4 then recruits the karyopherin Kap104, for import into the nucleus. Importantly, a crystal structure is available for Rpl4–Acl4 (F.H. and A.H., submitted), and the binding site of Kap104 on the complex has been mapped ([*Stelter et al., 2015*]; F.H. and A.H., submitted). In our GlyGly profiling efforts, measurements were obtained for three ubiquitination sites on Rpl4: K55, K308, and K338 (10-, 8-, and 1.6-fold decrease in *tom1Δ*, respectively) (*Supplementary file 3B*). The crystal structure of the ribosome indicates that K55 and K308, whose ubiquitinations exhibited the strongest dependence on Tom1, contact rRNA and are not accessible for modification in the mature ribosome (*Ben-Shem et al., 2011*). Interestingly, the crystal structure of the Acl4–Rpl4 complex revealed that Acl4 conceals K55, and a structural model of Kap104 docked

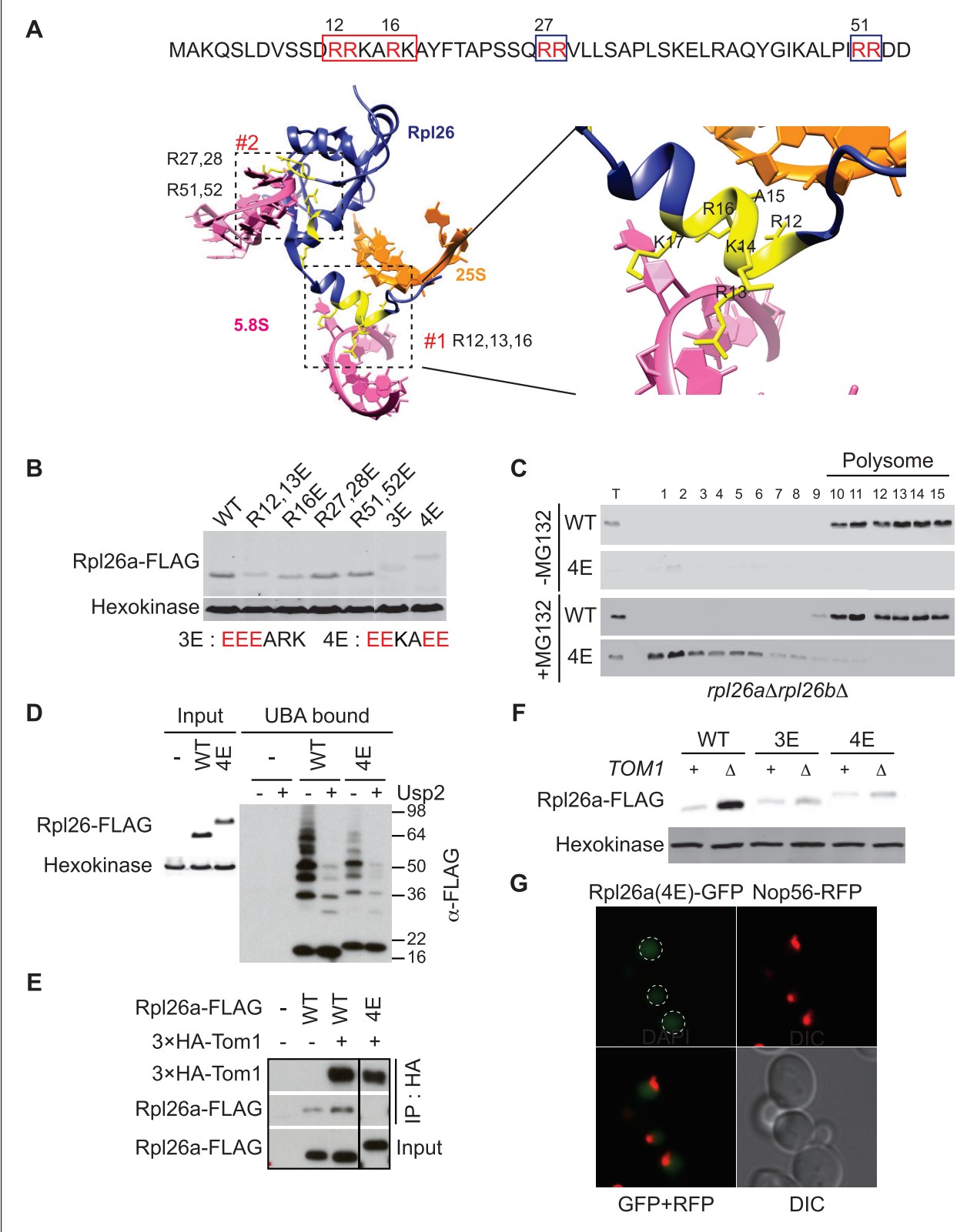

**Figure 4.** A short stretch of positively-charged residues in Rpl26a that mediates rRNA binding promotes association with Tom1. (**A**) Top: The first 54 amino acids of Rpl26a. Sequences adjacent to rRNA are boxed. Arginine residues targeted for mutation are in red. Bottom: Relative positions of arginines and rRNA based on the atomic model of the yeast 80S ribosome (PDB files 3U5D and 3U5E). Orange and pink ribbons correspond to 25S and 5.8S rRNA, respectively. Blue ribbon corresponds to Rpl26. Predicted regions (#1 and #2) for rRNA binding are highlighted in yellow and boxed. (**B**)

*Figure 4 continued on next page*

*Figure 4 continued*

Differential accumulation of WT and mutant Rpl26a[FLAG] upon galactose induction. n = 2 biological replicates. (C) Top: Ribosome assembly of WT Rpl26a[FLAG] or Rpl26a-4E[FLAG] induced in *rpl26aΔrpl26bΔ* cells. Bottom: Same as above except that MG132 was added 30 min after addition of galactose. T indicates total extract. n = 2 biological replicates. (D) Polyubiquitination of Rpl26a-4E[FLAG]. Assay was performed as described for **Figure 1D**. Samples in '+' lanes were treated with deubiquitinating enzyme Usp2 prior to processing for SDS-PAGE, to demonstrate that high MW species were modified with ubiquitin. n = 2 biological replicates. (E) The Rpl26-4E mutation disrupts binding to Tom1. Lysates from cells of the indicated genotypes were subjected to pull-down with anti-HA followed by SDS-PAGE and immunoblotting for the indicated proteins. n = 2 biological replicates. (F) Protein level of Rpl26a[FLAG] mutants upon galactose induction in WT and *tom1Δ* cells. n = 2 biological replicates. (G) Fluorescence images of Rpl26a[4E]-GFP induced in WT cells. Nop56-RFP marks nucleoli. Dashed circles indicate nuclear region as judged by DAPI staining. n = 2 biological replicates.

The following figure supplement is available for figure 4:

**Figure supplement 1.** Rpl26a-4E mutant is unstable and degraded by Doa10 in the nucleus/nucleolus.

to Acl4–Rpl4 indicates that it protects K308 and K338. Upon import of Rpl4–Acl4–Kap104 into the nucleus, Kap104 is dissociated through the action of Ran-GTP (**Kressler et al., 2012**)(F. H. and A. H., submitted). To test if this exposes the C-terminal region of Rpl4 to Tom1 (as would be the case if assembly of Rpl4 was delayed following nuclear import and release of Kap104), we performed an in vitro ubiquitination assay with purified substrates. [3xHA]Tom1 immunoprecipitates readily ubiquitinated Rpl4 in binary Acl4–Rpl4 complexes but not in ternary Acl4–Rpl4–ctKap104 complexes, despite the fact that at least eight lysines of Rpl4 should remain exposed in the ternary complex (red circle, **Figure 5A**). Ubiquitination of Rpl4 within the binary complex required its extended C-terminus because it was eliminated when the C-terminal region was truncated at residue 276 (**Figure 5B**). These data, along with those on Rpl26a, suggest that Tom1 selectively recognizes and ubiquitinates ribosomal proteins via residues that are only accessible in the unassembled state. Notably, pulse-chase labeling of yeast cells revealed that Rpl4Δ63–87, which lacks the loop region that binds Acl4, transiently associates with Tom1 (see **Figure 3B** of **Stelter et al., 2015**). However, it is unclear if loss of Tom1 during the chase was due to degradation or incorporation of Rpl4Δ63–87 molecules into ribosomes.

To assess more generally if Tom1 targets lysines that are inaccessible in the mature ribosome (pdb 4V88), we examined the disposition of the major Tom1-dependent ubiquitination sites on large subunit proteins reported in **Figure 3E**. For this analysis, we used the structure of the HECT domain of Rsp5 covalently conjugated to both a donor ubiquitin and a substrate acceptor (pdb 4LCD). We asked whether the epsilon amino group of a given lysine within the mature large subunit could conceivably make contact with the active site cysteine of Rsp5. Of the 18 lysines that could be observed in the 4V88 structure, 13 in the free 60S and 15 in the assembled 80S were not accessible to the probe (**Figure 5C**). Taken together, our data suggest strongly that Tom1-dependent ubiquitination events generally occur on ribosomal proteins prior to their assembly into the ribosome, on residues that normally are either buried in the ribosome, engage in salt bridges, or are otherwise shielded from contact.

## Tom1 is required for maintaining cellular homeostasis of ribosomal proteins

We next turned our attention to the phenotypic effects of Tom1 deficiency. If a limited capacity to degrade excess ribosomal proteins contributes to the temperature-sensitive growth defect of *tom1* mutants (**Utsugi et al., 1999**) (**Figure 1—figure supplement 2C and D**), we reasoned that conditions that foster imbalances in the production of ribosome components should exacerbate this defect. To test this, we performed three different perturbations. First, *tom1[CA]* cells (but not WT cells) were extremely sensitive to constitutive overexpression of *RPL26A* (**Figure 6A**) but not *RPL26A-4E* (**Figure 6—figure supplement 1A**) from the *GAL10* promoter. Second, we created a situation in which ribosomal proteins as a group are made in excess of rRNA via depletion (using the auxin-inducible degron (AID) [**Morawska and Ulrich, 2013**]) of proteins involved in rRNA synthesis including Rrn3 (transcription factor for RNA polymerase I), Rpa190 (RNA polymerase I largest subunit) and Hmo1 (regulator of transcription by RNA polymerase I). Each of these depletions caused a

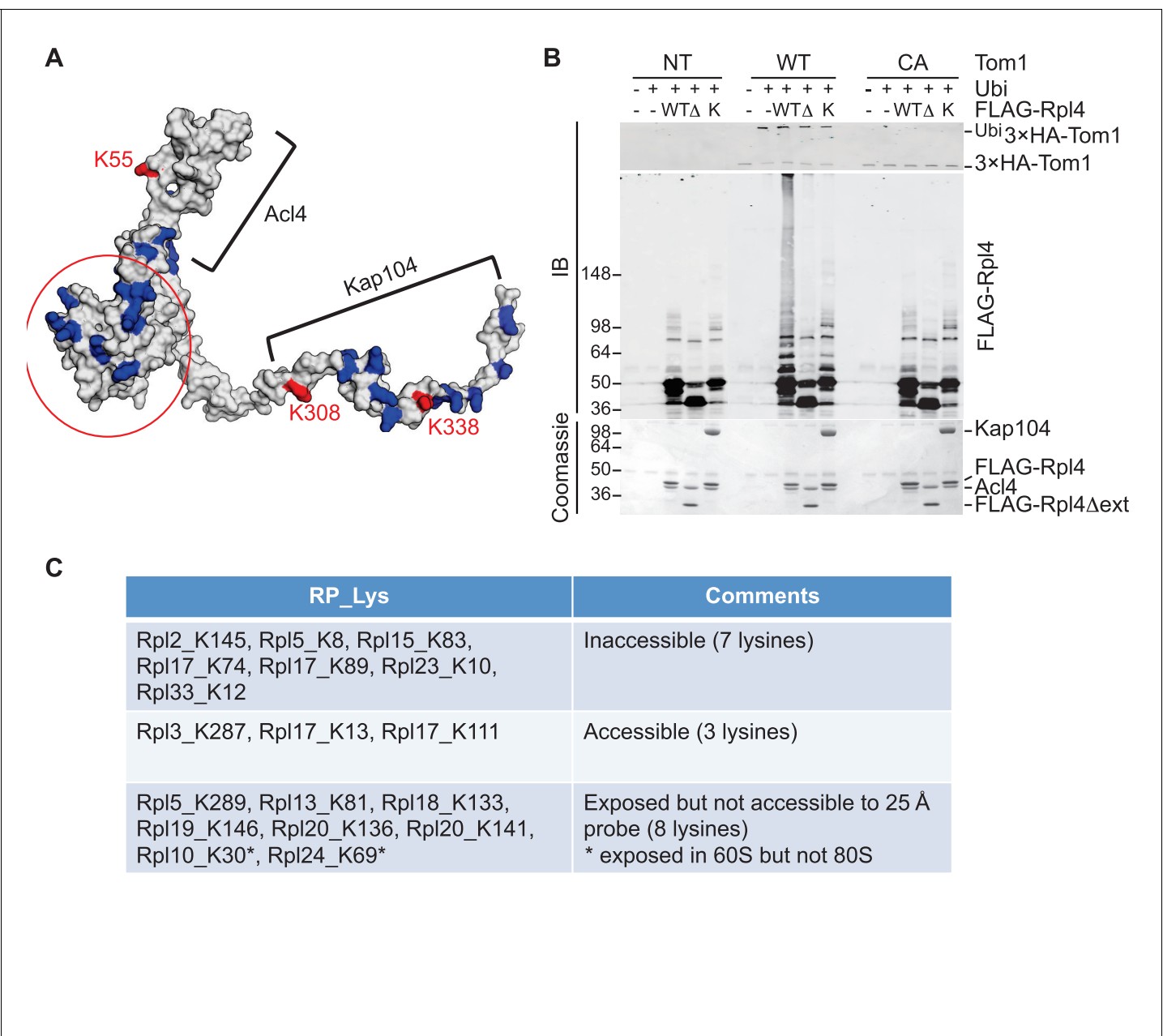

**Figure 5.** Tom1 acts through residues that are normally inaccessible in the structure of the mature ribosome. (**A**) Structure of Rpl4 within the mature ribosome (PDB ID 4V88). Lysine residues are colored blue, with K55, K308, and K338 colored red. Areas involved in binding Acl4 and *ct*Kap104 are indicated. The exact boundaries of the Kap104 binding site are not known. The globular central domain, which is fully exposed in the ternary Acl4–Rpl4–Kap104 complex but is not ubiquitinated, is circled in red. (**B**) Ubiquitination of Acl4–Rpl4 by [3xHA]Tom1. Anti-HA immunoprecipitates from untagged (NT), [3xHA]*TOM1* (WT), and [3xHA]*tom1*[CA] (CA) cells were supplemented or not with E1/E2/ubiquitin/ATP (Ubi) and purified Acl4–[FLAG]Rpl4, Acl4-[FLAG]Rpl4Δext and Acl4–[FLAG]Rpl4–*ct*Kap104 proteins. Samples were analyzed by SDS-PAGE and staining with Coomassie blue or immunoblotting with the indicated antibodies. WT, Δ, and K refer to Acl4–[FLAG]Rpl4, Acl4-[FLAG]Rpl4Δext and Acl4–[FLAG]Rpl4–*ct*Kap104, respectively. See detailed methods in Material and methods. n = 3 biological replicates. (**C**) Tom1 preferentially targets lysines that are inaccessible in mature ribosomes. Lysine residues shown are those from large subunit ribosomal proteins in *Figure 3E* that are incorporated in the model for the structure of the yeast ribosome (pdb 4V88). The structure of a HECT domain–donor ubiquitin complex (pdb 4LCD) predicts that a gap of radius 25 Å must be present for Tom1 to access a lysine for ubiquitination. Two of the sites (Rpl10 K30 and Rpl24 K69) are accessible in the 60S large subunit but become inaccessible upon formation of the 80S ribosome.

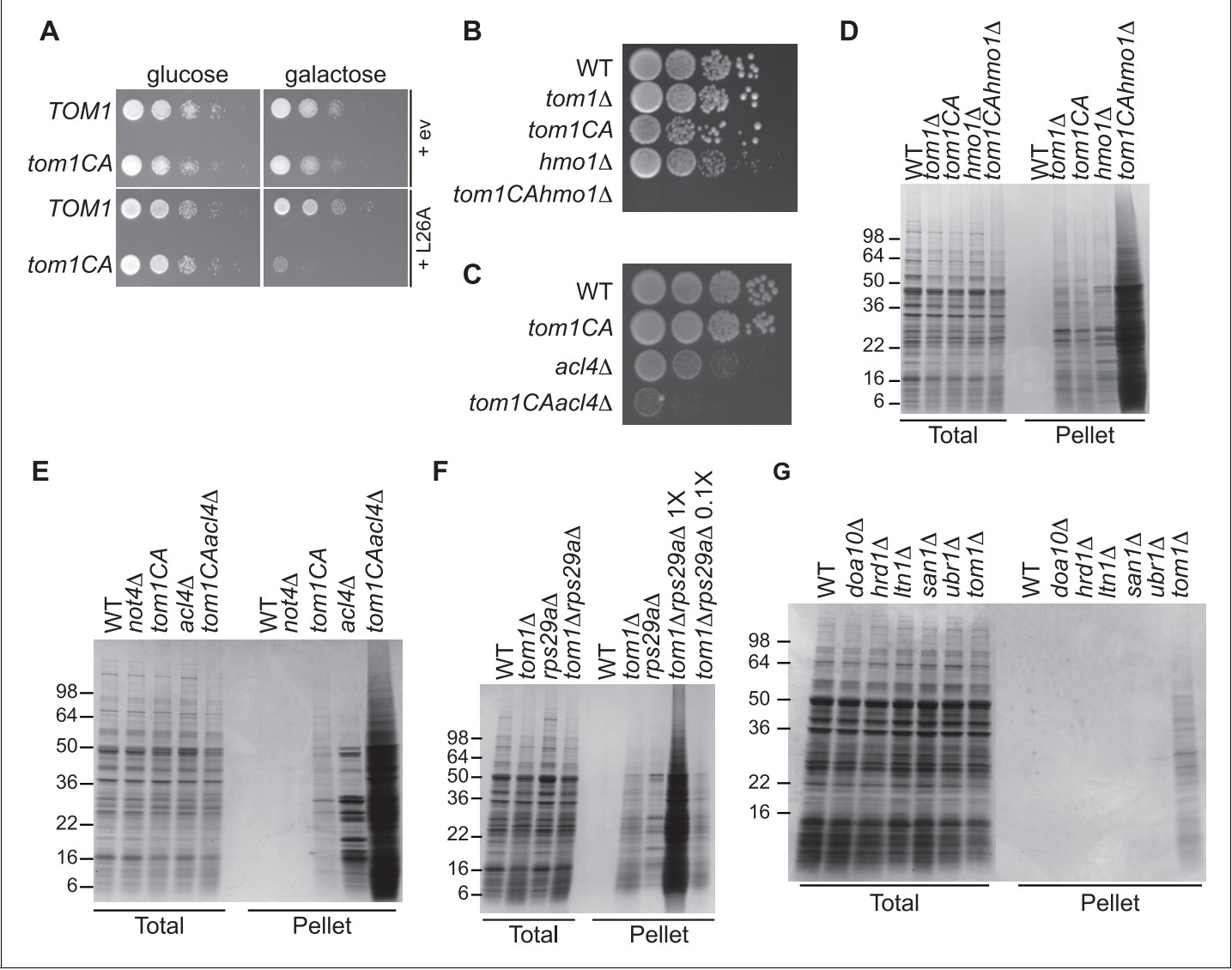

**Figure 6.** Defective ribosome assembly homeostasis and proteostatic collapse in *tom1* mutant cells. (**A–C**) Hypersensitivity of *tom1^CA^* cells to imbalances in ribosome components. (**A**) Cells of the indicated genotypes were spotted in serial 10-fold dilutions on glucose or galactose medium and incubated at 30°C for 2 days. ev refers to empty vector. n = 2 biological replicates. (**B, C**) As in (**A**) except that cells of the indicated genotypes were spotted on YPD. n = 2 biological replicates. (**D–G**) Massive accumulation of insoluble proteins in *tom1* mutant cells. Cells of the indicated genotypes were lysed and fractionated into detergent-soluble and insoluble fractions, which were separated by SDS-PAGE and stained with Coomassie Blue. The pellet fraction is overloaded 20-fold compared to the total and supernatant fractions. n = 2 biological replicates.

The following figure supplement is available for figure 6:

**Figure supplement 1.** Tom1 is required for maintaining proteostasis.

synthetic growth defect when combined with *tom1^CA^* (***Figure 6—figure supplement 1B***). The effect of combining *hmo1Δ* and *tom1^CA^* mutations was even more severe (***Figure 6B***). Third, we manipulated cells such that assembly of a single ribosomal protein was impaired, via deletion of the Rpl4-selective chaperone Acl4 (***Stelter et al., 2015***). Inactivation of *TOM1* in an *acl4Δ* background caused a substantial synthetic growth defect (***Figure 6C***).

If the synthetic growth defects described above arose from a catastrophic failure of proteostasis, we reasoned that this might manifest itself in the detergent-insoluble fractions of double mutant cells. Strikingly, there was a massive increase in detergent-insoluble proteins when RNA Pol I

transcription was diminished (*hmo1Δtom1^CA*; *Figure 6D*) or Rpl4 assembly was perturbed (*acl4Δtom1^CA*; *Figure 6E*) in a *tom1^CA* background.

Identification of Tom1 as a key mediator of ERISQ may rationalize numerous genetic interactions that have been reported for *tom1Δ*. In addition to suppressors that map to stress response/chaperone pathways and genes involved in ribosome protein expression (PKA pathway), deletions in 36 different genes encoding ribosomal proteins exhibit synthetic negative genetic interaction with *tom1Δ* (*Costanzo et al., 2010*) (*Figure 6—figure supplement 1C*). Based on the effects of the *acl4Δ* mutant (*Figures 6C,E*), deletions of one copy of duplicated ribosomal protein genes are predicted to create an imbalance in ribosomal proteins resulting in severe growth and proteostasis defects in a *tom1* background. Consistent with this prediction, deletion of one of the two copies of *RPS29* led to a synthetic growth defect (*Figure 6—figure supplement 1D*) and enormous accumulation of insoluble proteins in a *tom1Δ* mutant (*Figure 6F*).

Given the major role of ribosome production in the cellular economy, we evaluated the relative impact of the Tom1-dependent ERISQ pathway on overall proteostasis by comparing the amount of insoluble proteins in well-characterized PQC mutants including *doa10Δ* and *hrd1Δ* (ERAD; ER-associated degradation; [*Vembar and Brodsky, 2008*]), *ltn1Δ* (RQC; ribosome bound QC; [*Bengtson and Joazeiro, 2010*; *Brandman et al., 2012*; *Defenouillere et al., 2013*; *Verma et al., 2013*]), *san1Δ* (nuclear PQC; [*Gardner et al., 2005*]) and *ubr1Δ* (ERAD and cytoplasmic PQC; [*Eisele and Wolf, 2008*; *Heck et al., 2010*]). Under unperturbed conditions, *tom1Δ* cells contained significantly greater amounts of insoluble proteins compared to these other PQC mutants (*Figure 6G*, Pellet), suggesting that ERISQ is a major player in maintaining proteostasis in yeast.

## ERISQ pathway is conserved

A prior study in human cells demonstrated that a significant fraction of newly-synthesized ribosomal proteins imported into the nucleus is degraded by the UPS (*Lam et al., 2007*), suggesting that a PQC mechanism for unassembled ribosomal proteins is conserved in higher eukaryotes. To test this possibility, we evaluated transient expression of human Rpl26^FLAG (hRpl26^FLAG) in T-REx-293 cells treated with or without MG132 or bortezomib. Cells treated with these proteasome inhibitors accumulated greater amounts of overexpressed hRpl26^FLAG (*Figure 7A*, *Figure 7—source data 1*), consistent with the existence of a UPS pathway that degrades overexpressed ribosomal proteins. The closest human homolog of Tom1 is Huwe1. Knockdown of Huwe1 by shRNA (*Thompson et al., 2014*) in both T-REx-293 and HeLa cells (*Figure 7B*, *Figure 7—source data 2*) and knockout of *HUWE1* in HEK293T cells (*Choe et al., 2016*) (*Figure 7C*, *Figure 7—source data 3*) enabled transient overexpression of hRpl26^FLAG. Importantly, a cycloheximide chase experiment indicated that hRpl26^FLAG overexpressed in *HUWE1* knockout cells was stable (*Figure 7D*, *Figure 7—source data 4*).

To test if Huwe1 was required for ubiquitination of transiently overexpressed hRpl26^FLAG, control or Huwe1-depleted T-REx cells were co-transfected with plasmids encoding hRpl26^FLAG and ^HAubiquitin and then treated with MG132 to induce accumulation of ubiquitin conjugates. IP/Western blot analysis performed under denaturing conditions revealed that hRpl26^FLAG was modified by ^HAubiquitin in control but not Huwe1-depleted cells (*Figure 7E*). Consistent with this result, prior analysis of Huwe1-deficient cells by GlyGly profiling revealed reductions in the ubiquitination of multiple ribosomal proteins (*Thompson et al., 2014*).

## Discussion

Nearly 40 years ago Jonathan Warner and colleagues showed that multiple ribosomal proteins are rapidly degraded in both yeast and HeLa cells when ribosomal RNA synthesis is inhibited (*Gorenstein and Warner, 1977*; *Warner, 1977*). They and others went on to show that individual ribosomal proteins cannot be overexpressed from plasmids in yeast because the excess protein is rapidly degraded (*Abovich et al., 1985*; *Warner et al., 1985*). Essentially no progress was made towards understanding the underlying mechanism, until it was shown that ribosomal proteins are among the most abundant ubiquitin conjugates that accumulate in yeast cells with reduced proteasome activity (*Mayor et al., 2007*, *2005*), and that newly-synthesized ribosomal proteins are degraded by the proteasome in human nucleoli (*Lam et al., 2007*). More recently, we have shown that ribosome proteins produced in excess are ubiquitinated and then degraded in a proteasome-

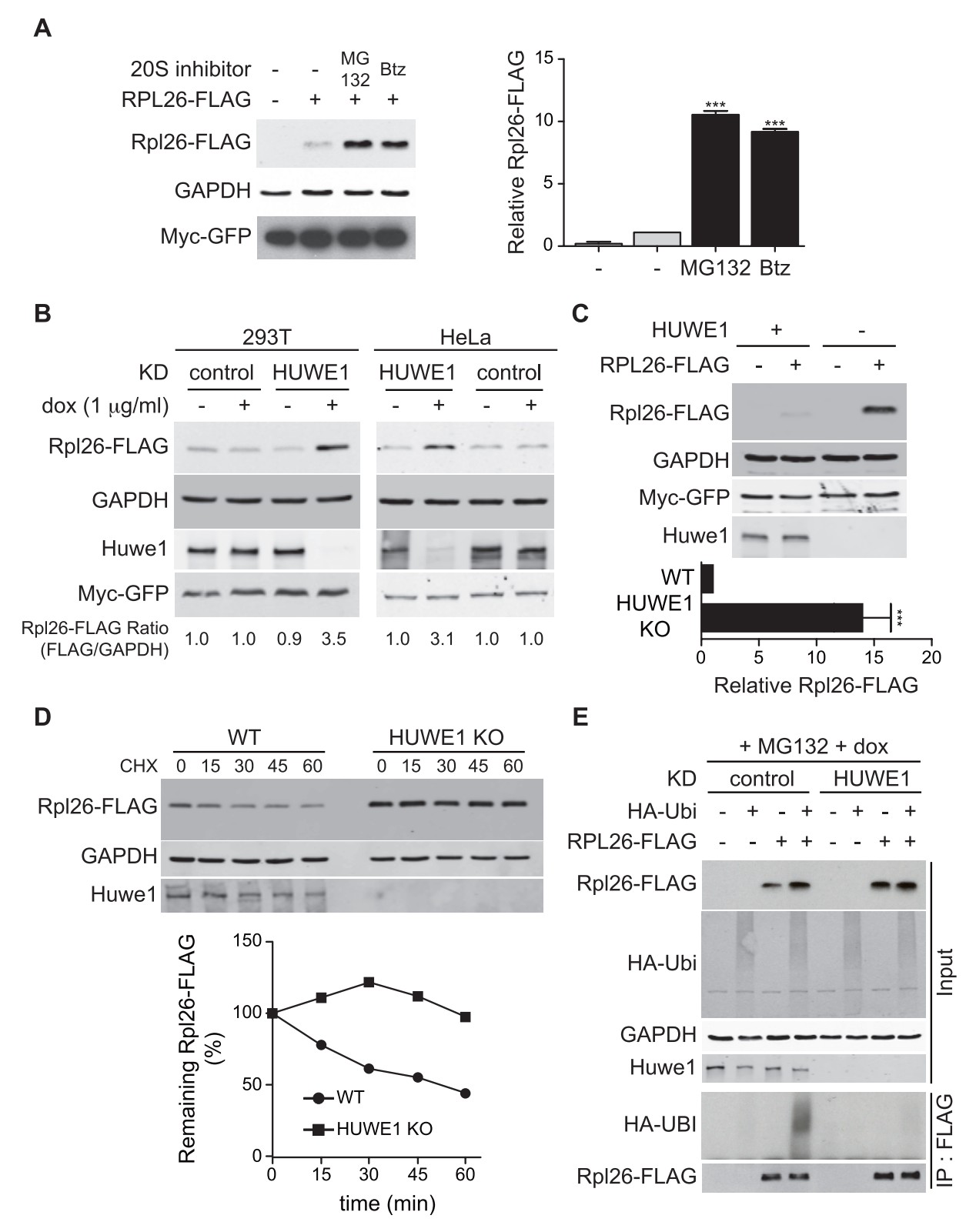

**Figure 7.** ERISQ is conserved in human cells. (**A**) Proteasome inhibition enables overexpression of human Rpl26. Left: transiently expressed hRpl26[FLAG] in T-REx-293 cells treated with 10 μM MG132 or 1 μM bortezomib (btz) for 3 hr. Right: quantification of blots. Values are the mean of three independent experiments and error bars indicate standard deviations. Asterisks indicate significant differences (two-tailed student's t-test, ***p<0.0001, compared with DMSO treatment). n = 3 biological replicates. Source data are available in *Figure 7—source data 1*. (**B**) Depletion of HUWE1 enables

*Figure 7 continued on next page*

*Figure 7 continued*
overexpression of human Rpl26. As in (A) except that T-REx-293 (left) and HeLa (right) cells were induced with doxycycline for 3 days to express stably integrated shControl or shHUWE1. The relative ratio of hRpl26$^{FLAG}$/GAPDH is shown below each lane. n = 3 biological replicates. Source data are available in *Figure 7—source data 2*. (C) Knockout of *HUWE1* enables overexpression of human Rpl26. Upper: as in (A) except that wild type and *HUWE1* knockout HEK293T cells were used. Bottom: quantification of blots. Values are the mean of three independent experiments and error bars indicate standard deviations. Asterisks indicate significant differences (two-tailed student's t-test, ***p<0.0001, compared with WT cells). n = 3 biological replicates. Source data are available in *Figure 7—source data 3*. (D) Overexpressed human Rpl26 is stable in *HUWE1* knockout cells. Upper: wild type and *HUWE1* knockout HEK293T cells transiently expressing hRpl26$^{FLAG}$ were treated with 10 µg/ml cycloheximide (CHX). Bottom: quantification of blot. n = 1 biological replicate. Source data are available in *Figure 7—source data 4*. (E) HUWE1 promotes ubiquitination of overexpressed human Rpl26. As in (B) except that $^{HA}$ubiquitin was co-expressed with Rpl26$^{FLAG}$ and MG132 was added 3 hr prior to cell lysis. Total cell extract prepared under denaturing condition was adsorbed to FLAG resin and the bound fraction was immunoblotted with antibodies against FLAG and HA. n = 2 biological replicates.

The following source data is available for figure 7:

**Source data 1.** Quantification of hRpl26-FLAG and GAPDH levels from three biological replicates.
**Source data 2.** Quantification of hRpl26-FLAG and GAPDH levels from one biological replicate.
**Source data 3.** Quantification of hRpl26-FLAG and GAPDH levels from three biological replicates.
**Source data 4.** Quantification of hRpl26-FLAG and GAPDH levels from one biological replicate.

dependent manner (*Sung et al., 2016*). In the work reported here, we shed additional light on this old puzzle by showing that the ubiquitin ligase Tom1 collaborates with the E2 enzymes Ubc4 and Ubc5 to broadly mediate degradation of unassembled ribosomal proteins in yeast. We refer to this pathway as ERISQ, for Excess Ribosomal protein Quality control.

## How are substrates recognized by the ERISQ pathway?

For ERISQ to work, two major challenges must be met. First, Tom1 has to detect many different ribosomal proteins. Second, it has to be able to distinguish their assembled versus unassembled forms. Although the exact structural basis for this discrimination remains to be determined, our analysis reveals that for Tom1 to act upon ribosomal proteins, residues that are normally concealed in the mature ribosome must be accessible. This was shown in vivo for the model substrate Rpl26 and in vitro for Rpl4. In addition, mapping of the major Tom1-dependent ubiquitination sites on the structure of the ribosome large subunit revealed that nearly 83% (15 of 18) of the sites on 12 different ribosomal proteins are no longer available to Tom1 following incorporation into the 80S ribosome. This suggests a very simple kinetic competition between Tom1 and rRNA for binding to ribosomal proteins newly arrived in the nucleolus. If the kinetic parameters for this race normally favor rRNA, the correct outcome would dominate and only those proteins that fail to assemble would be targeted.

Tom1 has also been implicated in the degradation of unassembled histones (*Singh et al., 2009*), in which case a similar kinetic competition could apply. Although the domain employed by Tom1 to bind its substrates remains unknown, we note that the previously-described Tom1 substrates Dia2 (*Kim and Koepp, 2012*), Hht2 (Singh et al., 2009) and Yra1 (*Iglesias et al., 2010*) have pI values ranging from 9.3–12. Thus, Tom1, which is overall acidic (pI 4.8) may have a negatively-charged region that interacts electrostatically with basic substrates.

## Conservation of the Tom1 eRISQ pathway in humans

Seminal studies from Warner indicated that human cells, like yeast, are unable to accumulate ribosomal proteins made in excess over rRNA due to rapid turnover (*Warner, 1977*), and subsequent proteomic studies revealed that a substantial fraction of newly-synthesized human ribosomal proteins are rapidly degraded by the proteasome (*Lam et al., 2007*). To test whether the mechanism we described in yeast also operates in human cells, we first demonstrated that the human ortholog (hRpl26) of yeast Rpl26 fails to accumulate upon transient overexpression, and then established that both ubiquitination and degradation of excess hRpl26 require Huwe1, which is the closest human

homolog of Tom1. A recent proteome-wide profiling of ubiquitination sites that exhibit diminished occupancy upon depletion of Huwe1 revealed that, excluding Huwe1 itself, six of the top ten affected sites and seventeen of fifty-six that exhibited ≥two-fold decreased occupancy were from ribosomal proteins (*Thompson et al., 2014*). These data point to a very general role for Huwe1 in ubiquitination of ribosomal proteins, similar to what we show here for yeast Tom1.

## Physiological significance of Tom1/Huwe1 and ERISQ

Cells lacking Tom1 exhibit a variety of phenotypes besides ERISQ including cell cycle arrest, nucleolar fragmentation, defective mRNA export from the nucleus, reduced Ngg1/Ada3-dependent transcription, reduced polysomes, reduced rate of rRNA processing/maturation (*Duncan et al., 2000*; *Saleh et al., 1998*; *Tabb et al., 2001*; *Utsugi et al., 1999*), and sensitivity to paromomycin (*Figure 6—figure supplement 1E*). On the one hand it is possible that all but one of these phenotypes are mainly secondary consequences of the primary defect. On the other hand, they may all arise independently from stabilization of different substrates. We can rule out that ERISQ is an indirect consequence of cell cycle arrest, nucleolar fragmentation, defective mRNA export, or a reduced rate of rRNA processing, because the defect in ERISQ is observed at temperatures that are permissive for growth of *tom1* mutants, whereas the other defects are only seen (or in the case of rRNA processing rate, is strongly enhanced) at the non-permissive temperature and thus are more likely to arise indirectly from exacerbation of a primary defect upon imposition of heat stress. It remains unclear to what extent a defect in ERISQ, when coupled to heat stress, might underlie these other phenotypes.

Some of the *tom1* phenotypes noted above are suggestive of a potential positive role for Tom1 in the assembly of functional ribosomes. However, this does not conflict with our observation that Tom1 plays a direct role in ubiquitinating and promoting the degradation of ribosome proteins that are overexpressed relative to their assembly partners. A speculative possibility is that Tom1 ubiquitinates a broad range of ribosomal proteins to promote their assembly (much as fusion to ubiquitin promotes assembly of Rps31; [*Finley et al., 1989*]), but if they fail to assemble within a given time interval, the ubiquitin conjugated by Tom1 serves to initiate degradation. However, the observation that genetic reductions of protein kinase A activity (which are predicted to diminish synthesis of ribosome proteins) suppress *tom1Δ* (*Figure 6—figure supplement 1C*) seems inconsistent with the idea that Tom1 plays a direct positive role in ribosome assembly.

A phenotype of *tom1$^{CA}$* cells that we describe here that is particularly notable is that they are exquisitely sensitive to perturbations that alter the balance between production of individual ribosomal proteins or total ribosomal proteins and rRNAs. Specifically, *tom1$^{CA}$* mutants are unable to sustain growth upon overexpression of *RPL26*, are sensitive to loss of the Rpl4-specific chaperone Acl4 (*Stelter et al., 2015*), are sensitive to reduction in expression (through deletion of one of two alleles) of a single ribosomal protein, and are hypersensitive to reduction of function in three different proteins involved in transcription by RNA polymerase I. Moreover, *tom1* mutants accumulate high levels of insoluble ribosomal proteins (which is consistent with their inability to degrade excess ribosomal proteins that may exist at any given point in time), and combining *tom1* with a mutation that causes an imbalance in ribosome production (e.g. *tom1$^{CA}$ acl4Δ*) leads to an enormous increase in insoluble protein, suggestive of a collapse in cellular proteostasis. Notably, even the *tom1Δ* single mutant accumulates more insoluble protein than any other PQC mutant that we examined, including the ERAD mutants *doa10Δ* and *hrd1Δ* (*Vembar and Brodsky, 2008*), the ribosome QC mutant *ltn1Δ* (*Bengtson and Joazeiro, 2010*; *Brandman et al., 2012*; *Defenouillere et al., 2013*; *Verma et al., 2013*), the nuclear QC mutant *san1Δ* (*Gardner et al., 2005*), or the cytosolic QC mutant *ubr1Δ* (*Eisele and Wolf, 2008*; *Heck et al., 2010*). Taken together with our observations that Tom1 directly binds and ubiquitylates unassembled ribosome proteins, these data point to a direct and critical role for Tom1 in the cellular proteostasis network.

The hypersensitivity of *tom1$^{CA}$* cells to stoichiometric imbalances in ribosome components raises interesting questions as to why unassembled ribosomal proteins would be toxic, and whether feedback mechanisms exist to monitor and respond to failures in ERISQ. Regarding the first question, given that ribosomal proteins are highly expressed, positively charged nucleic acid binding proteins, their accumulation might interfere with RNA biology. Regarding feedback mechanisms, it is interesting to note that in human cells, reduction of rRNA expression by low-dose actinomycin D treatment results in poor assembly of several ribosomal proteins including Rpl5 (*Dai and Lu, 2004*), Rpl11

(*Lohrum et al., 2003*; *Zhang et al., 2003*), Rpl26 (*Zhang et al., 2010*), Rps7 (*Chen et al., 2007*) and Rps14 (*Zhou et al., 2013*). These unassembled ribosomal proteins bind to and titrate ubiquitin ligase Mdm2, which leads to stabilization and accumulation of the Mdm2 substrate p53. This provides a sensitive feedback loop to reduce cell growth in response to stresses that impede ribosome assembly. It will be of great interest to determine whether a similar feedback mechanism operates in yeast, and how Huwe1 activity relates to the Mdm2–p53 feedback pathway described in human cells.

### Is ERISQ relevant to cancer?

Dysfunctional ribosomal proteins and ribosome biogenesis has been linked to many diseases (*Freed et al., 2010*; *Narla and Ebert, 2010*). Particularly, given that one of the characteristic features of cancer is an increase in the overall rate of protein synthesis, it is clear that regulation of ribosome biogenesis is closely associated with tumor progression. Consistent with this expectation, RNA Pol I activity is highly elevated in many cancer cells and it leads to the enlargement of the nucleolus, which has been used as a marker for cancer for over 100 years (*Derenzini et al., 2009*). Furthermore, two major human tumor suppressor proteins, pRB and p53, have been shown to repress the production of rRNA and the loss of these factors cause an up-regulation of ribosome biogenesis in cancer tissues, consistent with a close relationship between cancer and ribosome synthesis (*Montanaro et al., 2008*). Up-regulation of ribosome production in cancer cells implies an enhanced dependency on QC mechanisms that survey ribosome assembly. However, the molecular pathway that mediates ERISQ in human cells has, up to now, remained unknown. Interestingly, several observations link Huwe1 function to cancer. Huwe1 is overexpressed in multiple human tumors, is essential for proliferation of a subset of tumors (*Adhikary et al., 2005*), and is required for activation of Myc-inducible target genes including ribosomal proteins in colon carcinoma cells (*Peter et al., 2014*). Our data suggest the intriguing hypothesis that accumulation of unassembled ribosomal proteins would titrate Huwe1, resulting in reductions in both Myc activity and transcription of genes that encode ribosomal proteins. Insights into the detailed molecular basis underlying ERISQ afforded by the discovery reported here will enable investigations into the biological relevance of ERISQ in human pathologies, including cancer, which may lead to novel concepts for therapy.

## Materials and methods

### Yeast strains, growth conditions, and transformation

All yeast strains used in this study (listed in *Supplementary file 1*) were derived from BY4741 (*MAT*a *his3Δ1 leu2Δ0 met15Δ0 ura3Δ0*) or W303a (*MAT*a *leu2-3,112 trp1-1 can1-100 ura3-1 ade2-1 his3-11,15*). All transformants were verified by auxotrophic selection or genomic PCR. Yeast was grown at 30°C in YPD or appropriate synthetic complete (SC) drop-out media. For ectopic expression of proteins from the *GAL1,10* promoter, cells grown in raffinose containing medium were treated with 2% galactose for 60–90 min. We note that experiments were initiated with cells at $OD_{600}$ 3.0, because the ERISQ pathway was more prominent in cells at mid-log phase ($OD_{600}$ ~3.0) compared to early-log phase ($OD_{600}$ ~1.0) (data not shown). Yeast transformation was performed by lithium acetate method (*Gietz and Schiestl, 2007*). For several strains, PCR products were generated by the 'gene splicing by overlap extension' method (*Horton et al., 1989*).

### Mammalian cell culture and transfection

T-REx-293 (Thermo Fisher Scientific, Waltham, MA), HeLa (*Thompson et al., 2014*), HEK293 (*Choe et al., 2016*) cells were grown in Dulbecco's modified Eagle's medium (DMEM) supplemented with 10% fetal bovine serum (Atlanta Biologicals, Flowery Branch, GA), penicillin, and streptomycin (Invitrogen, Carlsbad, CA) at 37°C in 5% $CO_2$. Mycoplasma contamination has been tested negative by MycoAlert Mycoplasma detection kit (Lonza, Switzerland). Cells used in this study were not in the database of cross-contaminated or mis-identified curated by the International Cell Line Authentication Committee (ICLAC). We haven't authenticated cells by a third party. Transient transfections were performed using transfection reagents FuGENE HD (Promega, Madison, WI) according to manufacturer's instructions. For lentiviral production, T-REx-293 cell line (ThermoFisher) was transfected with the lentiviral construct along with helper plasmids. Forty eight hours after transfection, media supernatant containing the lentivirus was collected. The lentivirus-containing medium supplemented

with polybrene, was used to transduce the target cells. The doxycycline-inducible shRNA expression constructs (pLKO-Tet-ON vector (*Thompson et al., 2014*; *Wiederschain et al., 2009*)) containing the control (RDB3142; CAA CAA GAT GAA GAG CAC CAA) and shHUWE1 (RDB3143; TGC CGC AAT CCA GAC ATA TTC) sequences were used (*Thompson et al., 2014*). shHUWE1 described previously as si5635 (*Zhong et al., 2005*) was used. Transduced T-REx-293 cells transduced with the control or HUWE1 shRNA constructs were selected in the presence of 4 µg/ml of puromycin.

## Plasmids and yeast molecular genetic manipulations

All plasmids used in this study are listed in *Supplementary file 2*. To construct pESC(HIS)-P$_{GAL10}$-RPL26A(mutants)-FLAG used in *Figure 4*, site-directed mutagenesis was performed using Quik-Change Site-Directed Mutagenesis Kit (Agilent Technologies; 200519, Santa Clara, CA) according to manufacturer's instructions.

To construct N-terminally 3×HA-tagged Tom1, the ~1800 bp PCR product including the *KAN* selection marker and *RFA1* promoter was obtained using pKanMX6–P$_{RFA1}$–9Myc–AID* (*Morawska and Ulrich, 2013*) as a template, forward primer 5′-GAG AGG AAA AGA AGA AAA GGT AAA ACA ACG AAT ATT TTT CCG GAT CCC CGG GTT AAT TAA-3′ and reverse primer 5′-TCT TGT AAG TAT AAT CTG GTC TTC T-3′, and the ~180 bp PCR product encoding the 3×HA tag was obtained using pRS304-3×HA-TOM1 plasmid (*Duncan et al., 2000*) as a template, forward primer 5′- AGA AGA CCA GAT TAT ACT TAC AAG AAT GGA ATT CGG CCG CAT CTT TTA CC-3′ and reverse primer 5′- GTT TCT CCT TTC TTG CCT TTT CAC ACC GAG TAA AAA GCA CAG ATC TGC ACT GAG CAG CGT-3′. With the two PCR products as templates, the ~2000 bp PCR product was obtained using forward primer 5′- GAG AGG AAA AGA AGA AAA GGT AAA ACA ACG AAT ATT TTT CCG GAT CCC CGG GTT AAT TAA-3′ and reverse primer 5′- GTT TCT CCT TTC TTG CCT TTT CAC ACC GAG TAA AAA GCA CAG ATC TGC ACT GAG CAG CGT-3′. The obtained final PCR product was used for transformation, generating cells expressing $^{3×HA}$Tom1 from the *RFA1* promoter.

To construct *tom1$^{CA}$* mutants, the ~145 bp PCR product including the 3′ end region of *TOM1* with a C3235A mutation was obtained using pRS304-3×HA-TOM1$^{CA}$ plasmid (*Duncan et al., 2000*) as a template, forward primer 5′-TGA TTT TGG TTC ATC AGA AAG ACT ACC ATC ATC ACA TAC C-3′ and reverse primer 5′-CAA AAG CAG AGA GGC GCG CCT CAG GCA AGA CCA AAC CCT TCA TGC-3′, and the ~1700 bp PCR product including the *KlURA3* was obtained using pFA6a-GFP-KlURA3 plasmid (*Sung et al., 2008*) as a template, forward primer 5′-GCA TGA AGG GTT TGG TCT TGC CTG AGG CGC GCC TCT CTG CTT TTG-3′ and reverse primer 5′-CAT GGC GCT ATA ATT TA CAC GAA AAA TGA CGT CAT TGG TTC TGG AGG AAG TTT GAG-3′. With the two PCR products as templates, the ~1850 bp PCR product was obtained using forward primer 5′-TGA TTT TGG TTC ATC AGA AAG ACT ACC ATC ATC ACA TAC C-3′ and reverse primer 5′-CAT GGC GCT ATA AAT TTA CAC GAA AAA TGA CGT CAT TGG TTC TGG AGG AAG TTT GAG-3′. The obtained final PCR product was used for transformation, generating *tom1$^{CA}$* strains.

## Antibodies

Anti-Rpl1 and anti-Rpl3 were generous gifts from Jonathan Warner. Anti-FLAG (F1804; RRID:AB_262044; 1:10,000 dilution) was from Sigma (St. Louis, MO), anti-Hexokinase (H2035-02; 1:10,000 dilution) was from USBiological (Salem, MA), anti-HUWE1 (A300-486A; RRID: AB_2615536; 1:1,000 dilution) was from Bethyl laboratories (Montgomery, TX), anti-HA (SC-7392; RRID:AB_627809; 1:5,000 dilution) was from Santa Cruz (Dallas, TX), anti-myc (MMS-150R; RRID: AB_291325; 1:5,000 dilution) was from Covance (San Diego, CA), anti-Ubiquitin (05–944; RRID: AB_441944; 1:5,000 dilution) and anti-GAPDH (MAB374; RRID: AB_2107445; 1:5,000 dilution) were from EMD Millipore, and anti-His$_6$ (200-332-382; RRID: AB_10704645; 1:5,000 dilution) was from Rockland (Limerick, PA). For secondary antibody, HRP-conjugated anti-rabbit IgG (A6154; RRID: AB_258284; 1:10,000 dilution) and HRP-conjugated anti-mouse IgG (M8770; RRID: AB_260711; 1:10,000 dilution) were from Sigma, IR680RD conjugated anti-rabbit (926–68071; RRID: AB_10956166; 1:10,000 dilution) and IR800CW conjugated anti-mouse (926–32210; RRID: AB_621842; 1:10,000 dilution) were from LI-COR Biosciences (Lincoln, NE).

## Microscopic analysis

Yeast cells grown in raffinose-containing SC medium at 30°C (OD$_{600}$ ≤ 1.0) were induced with galactose for 1 hr to express Rpl26a$^{GFP}$ and placed in 384-well glass-bottom microplates (Whatman, UK) pretreated with concanavalin A (Sigma; L7647) to ensure cell adhesion. Fluorescence images were taken using a Zeiss Axiovert 200M Inverted Microscope with an FITC filter set (excitation band pass filter, 450–490 nm; beam splitter, 510 nm; emission band pass filter, 515–565 nm) and a Rhodamine filter set (excitation band pass filter, 546 nm; beam splitter, 580 nm; emission long pass filter, 590 nm). We analyzed at least 50 cells and subcellular localization of GFP-fused proteins was reconfirmed by co-localization assay as described previously (*Huh et al., 2003*).

## Immunoblotting

For denatured samples, yeast or mammalian cells were harvested, washed two times with PBS, and boiled in 2× SDS-containing sample buffer for 5 min followed by brief bead beating or sonication, respectively. For non-denatured samples, yeast cells were harvested and disrupted by bead beating in lysis buffer (50 mM Tris-HCl pH 7.5, 150 mM NaCl, 1% Triton X-100 andprotease inhibitor cocktail (EDTA-free; Roche, Switzerland)). Cell debris was removed by centrifuging at 3000 rpm for 5 min, and the remaining cell extract was centrifuged at 12,000 rpm for 10 min in an Eppendorf Centrifuge 5430R. For mammalian cells, harvested cells were washed twice with PBS, and then incubated with RIPA buffer (50 mM Tris-HCl pH 7.5, 150 mM NaCl, 1% IGEPAL, 0.5% sodium deoxycholate, 0.1% SDS and protease inhibitor cocktail (EDTA-free; Roche)) for 10 min. After centrifugation at 12,000 rpm for 10 min in an Eppendorf Centrifuge 5430R, the supernatant was transferred to a new tube and mixed with SDS-PAGE sample buffer. Hexokinase and GAPDH were used as an internal control. Myc-GFP was used as a control for transfection efficiency. Protein levels were quantified using Odyssey software.

## Protein expression and purification

Acl4-$^{FLAG}$Rpl4, Acl4-$^{FLAG}$Rpl4Δext and *Chaetomium thermophilum* Kap104 were expressed in *E. coli* BL21-CodonPlus(DE3)-RIL cells (Stratagene, San Diego, CA) grown in LB media supplemented with appropriate antibiotics. Protein expression was induced at an OD$_{600}$ of approximately 0.6 with 0.5 mM isopropyl β-D-thiogalactoside (IPTG) for ~18 hr at 18°C (Acl4-$^{FLAG}$Rpl4, Acl4-$^{FLAG}$Rpl4Δext) or 23°C (*ct*Kap104). Cells were harvested by centrifugation and resuspended in a buffer containing 20 mM Tris-Base pH 8.0, 500 mM NaCl, 5 mM β-mercaptoethanol (Sigma), 2 µM bovine lung aprotinin (Sigma), and complete EDTA-free protease inhibitor cocktail (Roche), and subsequently flash frozen in liquid nitrogen. Thawed cells were lysed with a cell disrupter (Avestin, Germany) and the lysate was centrifuged for 1 hr at 40,000 ×g. Cleared lysate of *ct*Kap104 expression was applied to a glutathione sepharose column equilibrated in buffer containing 20 mM Tris-Base pH 8.0, 100 mM NaCl, and 5 mM DTT (GE Healthcare, Pasadena, CA) and eluted via a glutathione gradient. Pooled fractions were cleaved with PreScission protease (GE Healthcare) for 12 hr. Cleared lysate of Acl4-$^{FLAG}$Rpl4, Acl4-$^{FLAG}$Rpl4Δext expression was applied to a Ni-NTA column equilibrated in buffer containing 20 mM Tris-Base pH 8.0, 500 mM NaCl, and 5 mM β-mercaptoethanol and eluted via an imidazole gradient. Pooled fractions were cleaved with ubiquitin-like-specific protease 1 (Ulp1) for 12 hr. Cleaved proteins were bound to a HiTrapQ HP (GE Healthcare) column equilibrated in buffer containing 20 mM Tris-Base pH 8.0, 100 mM NaCl, and 5 mM DTT and eluted via a linear NaCl gradient, concentrated, and injected onto a HiLoad 16/60 Superdex 200 column equilibrated in 20 mM Tris-Base pH 8.0, 100 mM NaCl, and 5 mM DTT. The Acl4-$^{FLAG}$Rpl4-*ct*Kap104 complex was assembled by stoichiometric incubation for 1 hr at 4°C and injection onto a HiLoad 16/60 Superdex 200 column equilibrated in 20 mM Tris-Base pH 8.0, 100 mM NaCl, and 5 mM DTT.

## In vitro ubiquitination assay

Five hundred OD$_{600}$ units of cells expressing $^{3×HA}$Tom1 (WT) and $^{3×HA}$Tom1$^{CA}$ were grown in raffinose medium and then induced to express Rpl26a$^{FLAG}$ in galactose medium for 45 min followed by bortezomib (50 µM) treatment for an additional 45 min. Cells were harvested and disrupted by bead beating in 2.5 ml lysis buffer (50 mM Tris-HCl pH 7.5, 150 mM NaCl, 0.5% NP-40 andprotease inhibitor cocktail (EDTA-free; Roche)). Cell debris was removed by centrifuging at 3,000 rpm for 5 min, and the remaining cell extract was centrifuged at 12,000 rpm for 10 min in an Eppendorf Centrifuge

5430R. Total lysates were incubated overnight with 150 µl anti-HA magnetic beads. Beads were washed three times with the same lysis buffer and then twice with 1× ubiquitin reaction buffer (50 mM Tris-HCl pH 8.0, 10 mM MgCl$_2$, 0.2 mM CaCl$_2$, 1 mM DTT and 5 µM MG132). Beads were incubated with 2 mM ATP, 166 nM E1 (Ube1; BostonBiochem; E-305), 1 µM E2 (UbcH5a; BostonBiochem; E2-616) and 20 µg of His$_6$-ubiquitin (Ubiquitin; BostonBiochem, C a m b r i d g e , M A; U-530) at 30°C for 1 hr. For Rpl4 ubiquitination, 10 µg of purified substrates (Acl4-$^{FLAG}$Rpl4, Acl4--$^{FLAG}$Rpl4Δext and Acl4-$^{FLAG}$Rpl4-ctKap104) were used. One-fifth volume of 5× SDS-containing sample buffer was added to stop the reactions and boiled for 5 min.

## Binding to UBA columns and Usp2 treatment

Immunoprecipitation of ubiquitin conjugates was performed as described with some modifications (*Verma et al., 2013*). TUBE2-UBA resin (BostonBiochem; AM-130) was used to bind polyubiquitinated substrates. One hundred OD$_{600}$ units of cells were harvested and disrupted by bead beating in 500 µl lysis buffer (20 mM Tris-HCl pH 7.5, 150 mM NaCl, 1 mM EDTA, 10% Glycerol, 5 mM NEM, 1% Triton X-100 andprotease inhibitor cocktail (EDTA-free; Roche)). Cell debris was removed by centrifuging at 3000 rpm for 5 min, and the remaining cell extract was centrifuged at 12,000 rpm for 10 min in an Eppendorf Centrifuge 5430R. TUBE2-UBA resin (30 µl) incubated overnight was washed three times with buffer (20 mM Tris-HCl pH 8.0, 150 mM NaCl, 0.5% Triton X-100). Fifty µl of 2× SDS-containing sample buffer was added to the resin and boiled for 5 min. Note that unmodified proteins can also potentially bind due to indirect interactions with ubiquitinated proteins. For Usp2 treatment, TUBE2-UBA resin prepared as described above was washed twice with 1× ubiquitin reaction buffer (50 mM Tris-HCl pH 8.0, 10 mM MgCl$_2$, 0.2 mM CaCl$_2$ and 1 mM DTT) and mixed with 1 µM Usp2 (BostonBiochem; E-504) at 30°C for 1 hr.

## Nuclear fractionation of yeast cells

Subcellular fractionation was performed as described (*Keogh et al., 2006*). One hundred OD$_{600}$ units of cells grown in rich medium (OD$_{600}$≤1.0) were collected by centrifugation and then treated with 200 units Zymolyase for 1 hr at 30°C in 1 ml SB buffer (1 M Sorbitol, 20 mM Tris-HCl pH 7.5, 10 mM β-mercaptoethanol). Spheroplasts collected by centrifugation (2000 rpm for 5 min at 4°C) were washed twice with SB buffer, and then resuspended in 500 µl EBX buffer (20 mM Tris-HCl pH 7.5, 100 mM NaCl, 0.25% Triton X-100, 15 mM β-mercaptoethanol and protease inhibitor cocktail (EDTA-free; Roche)). An aliquot was taken and used as a total cell extract, and the remainder of the lysate was layered over 1 ml NIB buffer (20 mM Tris-HCl pH 7.5, 100 mM NaCl, 1.2 M Sucrose, 15 mM β-mercaptoethanol and protease inhibitor cocktail (EDTA-free; Roche)) and centrifuged (12,000 rpm for 15 min at 4°C in an Eppendorf Centrifuge 5430R). A sample of the upper soluble fraction was taken and used as cytosol and the rest of the supernatant discarded. The glassy white nuclear pellet was suspended in 500 µl EBX buffer and kept on ice for 10 min with gentle mixing and an aliquot taken and used as the nuclear fraction. 2× SDS-PAGE loading buffer was added to each fraction and samples were incubated at 95°C for 5 min and then subjected to SDS-PAGE and Western analyses.

## Sucrose gradient and polysome profiling

Sucrose gradient and polysome profiling were performed as described (*Verma et al., 2013*). Yeast cells were grown to logarithmic phase in rich medium supplemented with glucose or raffinose at 30°C, and treated with cycloheximide (100 µg/ml) for 15 min before cell lysis to stabilize polysomes. One hundred OD$_{600}$ units of cells were harvested and disrupted by bead beating in lysis buffer (0.5 mM DTT, 100 µg/ml cycloheximide, 200 µg/ml heparin, 20 mM Tris-HCl pH 7.5, 140 mM KCl, 5 mM MgCl$_2$ andprotease inhibitor cocktail (EDTA-free; Roche)). Cell debris was removed by centrifuging at 3000 rpm for 5 min, and the remaining cell extract was centrifuged at 12,000 rpm for 10 min in an Eppendorf Centrifuge 5430R. Twenty five A$_{260}$ units of cell lysate layered on 7%~47% discontinuous sucrose gradient prepared in buffer (1 mM DTT, 140 mM KCl, 20 mM Tris-HCl pH 7.5 and 5 mM MgCl$_2$) were centrifuged in SW55Ti rotor for 90 min at 50,000 rpm. For polysome profiling analysis, samples were fractionated while continuously recording the absorbance at 254 nm with a UV detector (ISCO, Lincoln, NE). For Western blot, 0.2 ml fractions collected from the top were treated with 0.02% sodium deoxycholate for 30 min on ice and then precipitated by adding TCA to 10% final

concentration for 1 hr. Pellets were washed with ice-cold acetone, and then resuspended in 2× SDS-containing sample buffer.

## Isolation of protein aggregates

Isolation of protein aggregates from yeast cells was performed as described previously (*Koplin et al., 2010*) with slight modifications. One hundred OD$_{600}$ units of exponentially growing cells were harvested, and cell pellets were frozen in liquid N$_2$. The cell pellets were resuspended in 1 ml lysis buffer (20 mM Na-phosphate pH 6.8, 10 mM DTT, 1 mM EDTA, 0.1% Tween, 1 mM PMSF, protease inhibitor cocktail and 100 units/ml zymolyase) and incubated at 30° C for 30 min. Chilled samples were treated by tip sonication (20%, 10 sec, twice) and centrifuged for 20 min at 600 g at 4°C. Supernatants were adjusted to identical protein concentrations, and aggregated proteins were pelleted at 16,000 g for 20 min at 4°C. After removing supernatants, insoluble proteins were washed once with Wash I buffer (20 mM Na-phosphate pH 6.8, 500 mM NaCl, 5 mM EDTA, 2% NP-40, 1 mM PMSF, and protease inhibitor cocktail), and centrifuged at 16,000 g for 20 min at 4°C. Insoluble proteins were washed twice with Wash II buffer (20 mM Na-phosphate pH 6.8) and sonicated (10%, 10 s, twice) in 40 µl of Wash II buffer. Pellets were processed either as described below or solubilized in 10 µl SDS sample buffer. 1X of the total cell lysate (T) and soluble (S) fractions, and 20X of the isolated pellet (P) fraction were separated by SDS-PAGE, and analyzed by Coomassie Blue staining and immunoblotting. For the experiment in *Figure 3—figure supplement 3A*, the cells were lysed by agitation with glass beads as described by *Kaganovich et al., (2008)*, in the presence of 3 different lysis buffers: 1) 100 mM Tris-Cl, 1% Triton X-100, 150 mM KCl, 5 mM MgCl$_2$ and protease inhibitor; 2) 100 mM HEPES, 1% Triton X-100, 300 mM NaCl and protease inhibitor (*Lu et al., 2014*); 3) 30 mM HEPES, 0.5% Triton X-100, 150 mM NaCl, 1% glycerol, 1 mM DTT and protease inhibitor (*Kaganovich et al., 2008*).

## SILAC labeling of cells

In order to determine the effect of deleting *TOM1* on ubiquitination in vivo, stable isotope labeling of amino acids in cell culture (SILAC) experiments were performed. Wild type or *tom1Δ* yeast strains were grown in either heavy or light medium and mixed 1:1. Three biological replicates of each strain was grown, one of which was a label swap. Yeast cells were grown in complete synthetic medium supplemented with 2% dextrose and 20 mg/l of L-lysine and L-arginine. Yeast cells labelled 'heavy', were grown in L-$^{13}$C$_6$$^{15}$N$_2$-lysine (Lys 8) and L-$^{13}$C$_6$-arginine (Arg 6) (Cambridge Isotope Laboratories, T e w k s b u r y , M A), while yeast cells labelled 'light' were grown in L-lysine (Lys 0) and L-arginine (Arg 0). Yeast cells were allowed to grow for 10 generations to log-phase (OD$_{600}$ 0.6~1.0) in the appropriate media to ensure full incorporation of the desired labels. Incorporation was determined by LC-MS analysis of the derivatized amino acid hydrolysate as previously described (*Hess et al., 2002*). Cells were harvested by centrifugation at 5000 ×g for 5 min, washed twice with sterile water, flash frozen in liquid N$_2$ and stored at −80°C until lysis.

## Peptide preparation for mass spec

Cell lysis, digestion and peptide desalting procedure were followed according to the PTMScan Ubiquitin Remnant Motif (K-ε-GG) Kit #5562 from Cell Signaling Technology product manual with a few changes. Cells were lysed in 5 ml of lysis buffer (20 mM HEPES (pH 8.0), 9 M Urea, 1× Protease Inhibitor cocktail (Promega), and 1 mM PMSF). Yeast cells were lysed by vortexing in the presence of glass beads. Cells were vortexed 1 min with 1 min rests on ice in between, for 7 cycles. Lysate was cleared by centrifugation at 20,000 ×g for 15 min at 15°C, after which protein concentration was determined by Bradford. Cleared lysate was mixed 1:1, by using 10 mg of each label. The mixed sample was reduced for 45 min with 1.25 M DTT, by adding 1/278 (v/v) to achieve a final concentration of 4.48 mM DTT. Cysteine alkylation was performed by treating the lysate with 10 mM NEM, for 30 min at room temperature (RT) in the dark. Proteins were then digested with LysC (Wako) at a ratio of 1:200 for 4 hr at RT. Lysate was then diluted to 2 M urea by adding 50 mM Tris (pH 8.0). The partially digested lysate was subsequently digested with sequence grade trypsin (Promega) at a protein ratio of 1:100 in the presence of 1 mM CaCl$_2$ overnight (≥15 hr) at room temperature in the dark. The following morning the reaction was quenched by adding trifluoroacetic acid (TFA) to a final concentration of 0.1%. Insoluble material was removed from the digest by centrifugation at

4000 ×g for 15 min. Peptides were desalted using a 500 mg capacity Sep-pak column that initially was hydrated using 7 column volumes of ACN (21 ml), followed by an equilibration step with 7 column volumes of Buffer A (0.1% TFA in $H_2O$) (21 ml). Cleared peptides were loaded onto the resin by gravity flow, washed with 7 column volumes of Buffer A, followed by 3 column volumes of Wash buffer (0.1% TFA, 5% ACN in $H_2O$). Desalted peptides were eluted using 2 column volumes of Elution buffer (0.1% TFA, 40% ACN in $H_2O$) (6 ml). The resulting peptide sample was frozen by storing at −80°C for at least 1 hr then lyophilized to dryness.

## Basic reversed phase peptide fractionation

Peptide fractionation, cross-linking of antibody, and K-ε-GG peptide immunoprecipitation procedure was followed as in Udeshi et al., (*Udeshi et al., 2013a*) with a few changes. Briefly, dried peptides were resuspended in 1.5 ml of bRP buffer A (5 mM ammonium formate, pH 10.0 and 2% ACN v/v) and fractionated by HPLC using an Agilent Zorbax Extended $C_{18}$ 5 µM column (dimensions 9.4 × 250 mm). A 50 µl fraction of load was saved to analyze later as an MS detectable sample during nanoLC-MS/MS analysis and for purposes of protein normalization. The sample was fractionated using a step gradient with Buffer B (5 mM ammonium formate, pH 10.0 and 80% ACN v/v) as follows: 0–7.5 min (0–6% B), 7.5–10.5 min (6–8% B), 10.5–67.5 min (8–27% B), 67.5–73.5 min (27–31%), 73.5–79.5 min (31–39% B), 79.5–90 min (39–60%), and 90–116 min (60–80% B) all at a flow rate of 2.0 ml/min. Fractions were collected in one-minute increments and mixed non-contiguously into 5 bins, beginning with fraction 8 and ending with fraction 103. Binned fractions were frozen by storing at −80°C for more than 1 hr and subsequently lyophilized to dryness.

## K-ε-GG peptide antibody cross-linking and immunoprecipitation

In short, one aliquot of the K-ε-GlyGly peptide specific antibody (PTMScan Ubiquitin Remnant Motif (K-ε-GlyGly) Kit #5562, Limited Use License, Cell Signaling Technology) was washed with 3–1 ml aliquots of 100 mM sodium borate (pH 9.0) and cross-linked in the presence of 1 ml of DMP cross-linking solution (100 mM sodium borate, pH 8.0, 20 mM dimethyl pimelimidate, DMP) for 30 min at room temperature with gentle rotation. Antibody bound beads were pelleted after each wash by centrifugation at 2000 ×g for 30 s and kept on ice whenever possible. The cross-linking reaction was performed to decrease contamination of K-ε-GG peptides with antibody during the final peptide elution step. The cross-linking reaction was quenched by washing the beads with 3–1 ml aliquots of 200 mM ethanolamine blocking buffer (pH 8.0) followed by incubating the cross-linked antibody for 2 hr at 4°C with 1 ml of fresh ethanolamine blocking buffer. After blocking, the antibody-bound beads were washed with 3–1 ml aliquots of IAP buffer (50 mM MOPS, pH 7.2, 10 mM sodium phosphate, and 50 mM NaCl) and divided equally into 5 aliquots. Independently, the 5 desalted binned peptide samples were resuspended in 1.5 ml of 1× IAP buffer, the pH was measured (should be pH≅7), and cleared by spinning at maximum speed for 5 min. The peptide samples were allowed to incubate with the freshly cross-linked antibody for 2 hr at 4°C. After immunoprecipitation the beads were pelleted by centrifugation at 2000 × g for 1 min, resuspended in 500 µl of 1× IAP, and transferred to 0.67 ml tubes and washed 3 times with 500 µl of 1× IAP buffer. Following the IAP washes, the beads were washed once with 500 µl of 1× PBS, followed by one 500 µl of wash with mass spectrometry grade water (Fluka). Finally, the bound K-ε-GG peptides were eluted with 2 × 150 µl aliquots of 0.15% TFA. With each elution aliquot, the beads were incubated for 10 min at RT with periodic tapping on the tube to achieve proper mixing. The resulting eluents were combined, dried, desalted by HPLC using a Michrom Bioresources $C_{18}$ macrotrap, (Buffer A: 0.2% Formic Acid in $H_2O$; Buffer B: 0.2% Formic Acid in ACN) and concentrated *in vacuo*.

## NanoLC-MS/MS analysis

The dried immunoprecipitated peptides were resuspend in Buffer A (0.2% Formic Acid, 2% ACN, nanoLC grade 97.8% $H_2O$) and subjected to proteomic analysis using an EASY II nano-UPLC (Thermo Fisher Scientific) connected on-line to an Orbitrap Elite hybrid mass spectrometer with a nanoelectrospray ion source (Thermo Scientific) using settings similar to those previously described (*Porras-Yakushi et al., 2015*). Peptides were separated using a 15 cm silica analytical column with a 75 µm inner diameter packed in-house with reversed phase ReproSil-Pur $C_{18AQ}$ 3 µm resin (Dr Maisch GmbH, Amerbuch-Entringen, Germany). The flow rate was set to 350 nl/min, using a linear gradient

of 2%-32% B (0.2% Formic Acid, 80% ACN, 19.8% nanoLC grade $H_2O$). Mass spectrometry detectable samples were analyzed on a 159 min gradient, while basic reversed phase immunoprecipitated samples were analyzed on a 90 min gradient. The mass spectrometer was set to collect data in a data-dependent mode, switching automatically between full-scan MS and tandem MS acquisition. All samples were analyzed by ETD and decision tree fragmentation. For ETD fragmentation, the fifteen most intense precursor ions were selected, while the 20 most intense ions were selected for fragmentation using the decision tree method. Data acquisition was managed using Xcalibur 2.0.7 and Tune 2.4 software (Thermo Fisher Scientific).

## Mass spectrometry analyses of protein aggregates

Mass spectrometry analyses of protein aggregates were performed as described. Insoluble protein pellets were solubilized in an 8 M Urea buffer (40 mM Tris, 65 mM DTT, 100 mM Ammonium bicarbonate) containing cOmplete Protease Inhibitor Cocktail (Roche) and sonicated for 10 s at 10% of maximum amplitude using a Branson Digital Sonifier. Samples were digested and prepared for mass spectrometry as described in (*Pierce et al., 2013*). One hundred fifty ng of digested peptides from *tom1Δ* cells and equal volume of peptides from WT cells were analyzed using an EASY-nLC 1000 coupled to an Orbitrap Fusion. Spectra were analyzed by MaxQuant (v 1.5.3.30). Digested peptides were loaded onto a 26-cm analytical HPLC column (75 µm ID) packed in-house with ReproSil-Pur C18AQ 1.9 µm resin (120 Å pore size, Dr. Maisch, Ammerbuch, Germany). After loading, the peptides were separated with a 120 min gradient at a flow rate of 350 nl/min at 50°C (column heater) using the following gradient: 2–6% solvent B (7.5 min), 6–25% B (82.5 min), 25–40% B (30min), 40–100% B (1 min), and 100% B (9 min) where solvent A was 97.8% $H_2O$, 2% ACN, and 0.2% formic acid and solvent B was 19.8% $H_2O$, 80% ACN, and 0.2% formic acid. The Orbitrap Fusion was operated in data-dependent acquisition (DDA) mode to automatically switch between a full scan (*m/z*=350–1500) in the Orbitrap at 120,000 resolving power and a tandem mass spectrometry scan of Higher energy Collisional Dissociation (HCD) fragmentation detected in the ion trap (using TopSpeed). AGC target of the Orbitrap and ion trap was 400,000 and 10,000 respectively.

### Mass spec data analysis

Raw data was searched using MaxQuant (*Cox and Mann, 2008*; *Wagner et al., 2011*) (v 1.5.3.30) against the SGD yeast database (5911 entries) and a contaminant database (259 entries). Precursor mass tolerance was 4.5 ppm after automatic recalibration. Fragment ion tolerance was 0.5 Da. All default options were used except for SILAC diGly samples where the multiplicity was set to 2 with heavy labels Arg6 (+6.020129) and Lys8 (+8.014199). Tryptic digest was specified with up to two missed cleavages. Protein, peptide and site false discovery rates were less than 1% and were estimated using a target-decoy approach (*Elias and Gygi, 2010*). For non-GlyGly analysis, oxidation of methionine and protein N-terminal acetylation were specified as variable modifications and carbamidomethylation of cysteine was specified as a fixed modification. For GlyGly analysis, N-ethylmaleimide modification of cysteine (+125.0477) was specified as a fixed modification, while N-terminal acetylation (+42.0106), methionine oxidation (+15.9949), and the GlyGly remnant (+114.0429, not on peptide C-terminus with neutral losses of 57.0215 and 114.0429 to account for fragmentation in the GlyGly remnant) were set as variable modifications, as previously described (*Porras-Yakushi et al., 2015*). iBAQ protein quantitation and 'match between runs' were enabled.

For each pair of *tom1Δ* and WT samples, the difference between iBAQ abundances was used to identify the proteins that were most accumulated in the insoluble fraction in *tom1Δ*. The average difference between *tom1Δ* and WT samples across the three replicates was then used to identify the top 10 percent of proteins most accumulating. This set of 127 proteins was checked for annotation enrichment against all proteins identified in the sample using DAVID (*Huang et al., 2009a*, *2009b*). The most enriched terms included GO FAT cellular component annotations 'ribosome' (p-value<9.3E-40) and 'mitochondrion' (p-value < 6.3E-2). The distribution of average ΔiBAQ values for all proteins and for proteins with those annotations is in *Figure 3H*. The distributions of average ΔiBAQ values for the large and small ribosomal subunits were compared with non-ribosomal proteins in *Figure 3—figure supplement 3C*. The individual ribosomal proteins ΔiBAQ values are displayed in *Figure 3—figure supplement 3D* with error bars representing the standard error of the mean (SEM).

## Acknowledgements

We are grateful to members of the Deshaies laboratory for helpful advice and Rati Verma for critical reading of this manuscript. We thank Jonathan Warner, Christine Guthrie, Sarah Luchansky, Ling Song, Won-Ki Huh, George Moldovan, and Helle Ulrich for generously providing reagents and helpful discussions. We also thank Thang Nguyen and Rati Verma for numerous helpful suggestions regarding biochemical experiments. We thank Annie Moradian and Roxana Eggleston-Rangel for mass spectrometric assistance. RJD is an Investigator of the HHMI.

## Additional information

### Competing interests

RJD: Reviewing Editor, *eLife*. The other authors declare that no competing interests exist.

### Funding

| Funder | Grant reference number | Author |
| --- | --- | --- |
| Gordon and Betty Moore Foundation | GBMF775 | Tanya R Porras-Yakushi<br>Michael J Sweredoski<br>Sonja Hess |
| National Institutes of Health | F32GM112308 | Justin M Reitsma |
| Boehringer Ingelheim Fonds | | Ferdinand M Huber |
| V Foundation for Cancer Research | Albert Wyrick V Scholar Award | André Hoelz |
| Sidney Kimmel Foundation for Cancer Research | Scholar Award | André Hoelz |
| Camille and Henry Dreyfus Foundation | Teacher-Scholar Award | André Hoelz |
| Howard Hughes Medical Institute | | Raymond J Deshaies |
| Beckman Institute, California Institute of Technology | | Tanya R Porras-Yakushi<br>Michael J Sweredoski<br>Sonja Hess |
| Heritage Research Institute | | André Hoelz |
| Edward Mallinckrodt, Jr. Foundation | Scholar Award | André Hoelz |
| Donald E. and Delia B. Baxter Foundation | | Min-Kyung Sung |
| National Institutes of Health | 1S10RR029594 | Sonja Hess |

The funders had no role in study design, data collection and interpretation, or the decision to submit the work for publication.

### Author contributions

M-KS, Performed all experiments except for mass spectrometry analyses, Designed and interpreted experiments and drafted and edited manuscript; TRP-Y, Performed GlyGly mass spec analysis; JMR, Performed mass spec analysis of insoluble pellets; FMH, Prepared Rpl4 protein complexes for in vitro ubiquitination; MJS, Performed bioinformatic analysis of mass spec data; AH, Oversaw FMH and edited manuscript; SH, Oversaw analysis and interpretation of mass spec data and edited manuscript; RJD, Managed the project and participated in the design and interpretation of the experiments, Drafted and edited the manuscript

### Author ORCIDs

Min-Kyung Sung, http://orcid.org/0000-0002-0513-6834
Raymond J Deshaies, http://orcid.org/0000-0002-3671-9354

## Additional files

**Supplementary files**

• Supplementary file 1. Yeast strains used in this study.

• Supplementary file 2. Plasmids used in this study.

• Supplementary file 3. (A) Tom1 interactors identified by MS. (B) Dataset from SILAC-GlyGly analysis. (C) Insoluble proteins identified in *tom1* mutants.

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
