## [Decision Letter]

Thank you for submitting your article "A conserved quality-control pathway that mediates degradation of unassembled ribosomal proteins" for consideration by *eLife*. Your article has been reviewed by two peer reviewers, including Davis Ng (Reviewer #2), and the evaluation has been overseen by Wade Harper as the Reviewing Editor and Randy Schekman as the Senior Editor.

The reviewers have discussed the reviews with one another and the Reviewing Editor has drafted this decision to help you prepare a revised submission.

Summary:

Ribosomes represent a major site of protein quality control for newly synthesized proteins, and multiple mechanisms have been described for identifying various types of misfolded or incorrectly translated proteins emerging from the ribosome. By comparison, much less is known concerning quality control mechanisms for formation of the ribosome itself. It was initially found in 1977 by Warner that reduced ribosomal RNA expression did not affect the rate of production of many ribosomal proteins but that the "excess" ribosomal proteins were rapidly degraded. At the time, the mechanism of turnover of excess ribosomal proteins wasn't addressed and in fact, very little is known about how excess ribosomal proteins are degraded. This is in spite of the fact that production (and rapid turnover) of supernumerary ribosomal proteins has been linked with aneuploidy by the Amon lab and may also be a consequence of MYC overproduction for example.

This manuscript now identifies a major ubiquitin ligase in budding yeast responsible for turnover of excess ribosomal proteins as the HECT E3 TOM1. Using cells engineered to overexpress a ribosomal protein, the authors screened for non-essential E2s and E3s that stabilize the supernumerary ribosomal proteins – yielding TOM1 – and then performed a number of biochemical and genetic experiments to examine various aspects of the mechanism. Together with earlier work indicating degradation in the nucleus and the known location of TOM1 there, the model is that TOM1 is a factor that directly interacts with residues of ribosomal proteins that would normally be buried in an assembled ribosome. The authors also provide an initial series of experiments suggesting that HUWE1 is a potential ortholog for the E3 in mammals.

This is clearly and interesting and timely topic and both reviewers were pretty positive about the paper. Both reviewers felt that the evidence for the involvement of TOM1 in the process is clear. As indicated below, there were 4 major areas that the reviewers feel need to be addressed. These have to do, in part, to interpretation of the data and the possibility that other models for the basis of TOM1 function may not have been fully vetted. Some aspects of the study were somewhat unexplained or not entirely convincing, although in an initial study of a new quality control pathway, one would not expect everything to be resolved. It may be that some data already exists that will address the concerns and in some cases, specific controls or changes to the conclusions could address the issues.

We would very much like to see a revised version that addresses these issues.

Essential revisions:

1) Have the authors considered the possibility that the function of Tom1 is in ribosome assembly? This might explain its: 1) transient association in pulse-chase studies of Stelter et al. (Mol. Cell, 2015), 2) the preferential association with unassembled protein, 3) the major consequences of its deletion on protein homeostasis, 4) the accumulation of aggregated ribosomal proteins, 5) the synthetic genetic interactions with other assembly factors, and 6) the sensitivity to paromomycin. Indeed, the pattern of insoluble proteins in Figure 6 of a Tom1 mutant is very similar to a deficiency of a known ribosomal assembly factor (Acl4). The assembly factor idea might better explain the bias for 60S proteins in the ubiquitination and mass spectrometry data. This is probably worth discussing, and if there is clear reasons to favor the QC function, to articulate/show those.

2) The experimental conditions used to analyze protein aggregation may be sub-optimal. The authors have co-opted a protocol from a chaperone paper, but the buffer is not great if one is analyzing ribosomal proteins: it has very low salt, no Magnesium, and 1 mM EDTA. Thus, under these conditions, the integrity of ribosomes and assembly intermediates will possibly be compromised (due to Magnesium chelation), with the low salt perhaps favoring aggregation. It may therefore be possible that the aggregation phenotype is perhaps exaggerated, especially if the consequence of Tom1 is the accumulation of ribosome assembly intermediates (see point 1 above). One possible experiment would be to perform sucrose gradients (where the conditions are more suitable) using *tom1* deletion cells to more directly visualize and separate intact ribosomes/polysomes from aggregates so a reader can assess the extent of aggregation? If the authors are concerned about the large size of polysomes interfering with the analysis, the sample could be digested with nuclease first to convert everything to monosomes.

3) In Figure 6, it seems that an important control would be test whether overexpression of the 4E mutant of L26A (which is not a TOM1 client) would not be synthetic with tom1CA. In the absence of this, the effect might not be specific to an unassembled ribosomal protein, but just any misfolded protein.

4) In Figure 7, the authors test the HECT domain E3 HUWE1 for its role in supernumerary ribosomal protein turnover in mammalian cells. There are 2 issues with the data that make it sub-optimal. First, the RNAi experiments do not meet the typical standards of two or more shRNAs or a rescue experiment (which would obviously be challenging in this case). Second, the authors examine accumulation of Rpl26 with HUWE1 depletion but they don't look at turnover. The authors should perform turnover assays to definitely demonstrate an effect on turnover. This is important for completeness and to maintain appropriate standards. Alternatively, the reviewers think that the paper could stand on its own with just the yeast data.

---

## [Author Response]

*Essential revisions:*

*1) Have the authors considered the possibility that the function of Tom1 is in ribosome assembly? This might explain its: 1) transient association in pulse-chase studies of Stelter et al. (Mol. Cell, 2015), 2) the preferential association with unassembled protein, 3) the major consequences of its deletion on protein homeostasis, 4) the accumulation of aggregated ribosomal proteins, 5) the synthetic genetic interactions with other assembly factors, and 6) the sensitivity to paromomycin. Indeed, the pattern of insoluble proteins in Figure 6 of a Tom1 mutant is very similar to a deficiency of a known ribosomal assembly factor (Acl4). The assembly factor idea might better explain the bias for 60S proteins in the ubiquitination and mass spectrometry data. This is probably worth discussing, and if there is clear reasons to favor the QC function, to articulate/show those.*

This possibility was hotly debated within the lab. First of all we would like to note that there is not an appreciable bias towards 60S proteins in the insolubility assay (see Figure 3—figure supplement 3). The argument that Tom1 is involved in degradation of ribosomal proteins that do not assemble seems inescapable to us, based on the evidence provided. To recap: Tom1 associates with unassembled ribosomal proteins, it is required for their ubiquitination and degradation in cells, and it directly ubiquitinates them in vitro. This is pretty straightforward evidence that it is an ubiquitin ligase that degrades ribosomal proteins that fail to assemble, which is what we claim. What is less straightforward is whether it might be partly to blame for assembly failures in *tom1* cells; i.e. does Tom1 also have a chaperone function that promotes assembly of ribosomal proteins? We believe we can rule out an ubiquitination-independent chaperone function resides in Tom1, because the ligase-dead point mutant behaves much like the deletion mutant (although in some assays the deletion has a modestly greater defect which could be due to an E3-independent chaperone function).

That leaves one possibility, which is that Tom1 ubiquitinates ribosomal proteins as they arrive in the nucleolus, and this ubiquitination serves to facilitate assembly in some way. If assembly does not occur, perhaps the ubiquitin then serves to nucleate assembly of an ubiquitin chain, leading to degradation. According to this idea, Tom1 has a dual role in promoting assembly and degrading what does not assemble. It is an intriguing idea but difficult to test in a compelling manner. To test it in trans would require an extremely fast-acting conditional mutation that wipes out Tom1 function. Otherwise, in any comparison of wild type and *tom1Δ* cells, one would be left with the problem of how to untangle an observed defect with a direct role for Tom1 in chaperoning the process, versus an indirect effect of undegraded, unassembled ribosomal proteins causing interference. Unfortunately, such a mutant does not exist. The alternative would be to make a cis mutant in a ribosomal protein. However our analysis of two different ribosomal proteins suggested the sites of ubiquitination are, at least in the two cases shown, on residues that are normally concealed in the assembled structure (and hence potentially of functional significance, making their mutation problematic). Adding to this problem is that ribosomal proteins are quite rich in ubiquitination sites. To address this important comment, we have added the following text to the Discussion: “Some of the *tom1* phenotypes noted above are suggestive of a potential positive role for Tom1 in the assembly of functional ribosomes. […] However, the observation that genetic reductions of protein kinase A activity (which are predicted to diminish synthesis of ribosome proteins) suppress *tom1Δ* (Figure 6—figure supplement 1) seems inconsistent with the idea that Tom1 plays a direct positive role in ribosome assembly.”

*2) The experimental conditions used to analyze protein aggregation may be sub-optimal. The authors have co-opted a protocol from a chaperone paper, but the buffer is not great if one is analyzing ribosomal proteins: it has very low salt, no Magnesium, and 1 mM EDTA. Thus, under these conditions, the integrity of ribosomes and assembly intermediates will possibly be compromised (due to Magnesium chelation), with the low salt perhaps favoring aggregation. It may therefore be possible that the aggregation phenotype is perhaps exaggerated, especially if the consequence of Tom1 is the accumulation of ribosome assembly intermediates (see point 1 above). One possible experiment would be to perform sucrose gradients (where the conditions are more suitable) using tom1 deletion cells to more directly visualize and separate intact ribosomes/polysomes from aggregates so a reader can assess the extent of aggregation? If the authors are concerned about the large size of polysomes interfering with the analysis, the sample could be digested with nuclease first to convert everything to monosomes.*

We used low salt because the hydrophobic effect (which is thought to underlie most aggregation) is well-known to be exaggerated in high salt, and we were concerned that a high-salt buffer might exacerbate matters. Please note also that our method included a wash of the insoluble fraction with 500 mM salt to try to rule out aggregation of highly-charged proteins. Nevertheless, to address this comment more thoroughly we repeated an analysis of *tom1Δ* and wild type cells lysed in three different buffers: one that contains high salt and magnesium and two others that were used in previous reports (Kaganovich et al., 2008; Lu et al., 2014). In addition, we generated lysates by bead-beating of intact cells instead of detergent solubilization of spheroplasts. The new data are presented in Figure 3—figure supplement 3. The amount of protein in the pellet fraction is greater in these experiments than what was observed with our previous method. Two factors may account for this: either bead-beating causes more denaturation during cell lysis, and/or the omission of a pellet wash step in new protocol leads to greater contamination of the pellet fraction by soluble proteins. Regardless, the amount of protein in the pellet fraction is consistently greater in *tom1Δ* cells, indicating that the observations we made previously were not an artifact of the buffer or lysis conditions employed.

*3) In Figure 6, it seems that an important control would be test whether overexpression of the 4E mutant of L26A (which is not a TOM1 client) would not be synthetic with tom1CA. In the absence of this, the effect might not be specific to an unassembled ribosomal protein, but just any misfolded protein.*

To address this clever point, we repeated the spot assay shown in Figure 6 with *tom1CA* cells expressing Rpl26a-4E. Unlike Rpl26a-WT, Rpl26a- 4E did not cause growth inhibition of *tom1CA* cells when it was constitutively overexpressed from the *GAL10* promoter (Figure 6—figure supplement 1).

*4) In Figure 7, the authors test the HECT domain E3 HUWE1 for its role in supernumerary ribosomal protein turnover in mammalian cells. There are 2 issues with the data that make it sub-optimal. First, the RNAi experiments do not meet the typical standards of two or more shRNAs or a rescue experiment (which would obviously be challenging in this case). Second, the authors examine accumulation of Rpl26 with HUWE1 depletion but they don't look at turnover. The authors should perform turnover assays to definitely demonstrate an effect on turnover. This is important for completeness and to maintain appropriate standards. Alternatively, the reviewers think that the paper could stand on its own with just the yeast data.*

To address these valid points, we repeated the analysis shown in Figure 7 with *HUWE1* knockout cells, which were recently generated (Choe et al., 2016). Consistent with the results in Figure 7, hRpl26^FLAG^ made in excess accumulated in *HUWE1* knockout cells, while it remained at a low level in WT cells. In addition, we performed a cycloheximide-chase experiment, which showed that excess hRpl26FLAG in WT was unstable with a half-life of ~60 minutes, whereas the same protein was stabilized in *HUWE1* knockout cells. Overall, our additional data further supports the idea that Huwe1 mediated the ERISQ pathway in human cells.